**Variable contribution of wastewater treatment plant effluents to downstream nitrous**
**oxide concentrations and emissions**
Weiyi Tang[1, *], Jeff Talbott[2], Timothy Jones[2], Bess B. Ward[1]
Affiliations:
1. Department of Geosciences, Princeton University, Princeton, NJ 08544, USA
2. Department of Environmental Quality, Woodbridge, VA 22193, USA
[*]Correspondence to: weiyit@princeton.edu
**Abstract**
Nitrous oxide ($N_2O$), a potent greenhouse gas and ozone-destroying agent, is produced during
nitrogen transformations in both natural and human-constructed environments. Wastewater
treatment plants (WWTPs) produce and emit $N_2O$ into the atmosphere during the nitrogen removal
process. However, the impact of WWTPs on $N_2O$ emissions in downstream aquatic systems
remains poorly constrained. By measuring $N_2O$ concentrations at a monthly resolution over a year
in the Potomac River Estuary, a tributary of Chesapeake Bay in the eastern United States, we found
a strong seasonal variation in $N_2O$ concentrations and fluxes: $N_2O$ concentrations were larger in
fall and winter but the flux was larger in summer and fall. Observations at multiple stations across
the Potomac River Estuary revealed hotspots of $N_2O$ emissions downstream of WWTPs. $N_2O$
concentrations were higher at stations downstream of WWTPs compared to other stations (median:
21.2 nM vs 16.2 nM) despite the similar concentration of dissolved inorganic nitrogen, suggesting
the direct discharge of $N_2O$ from WWTPs into the aquatic system or a higher $N_2O$ production yield
in waters influenced by WWTPs. Meta-analysis of $N_2O$ measurements associated with WWTPs
globally revealed variable influence of WWTPs on downstream $N_2O$ concentrations and
emissions. Since wastewater production has increased substantially with the growing population
and is projected to continue to rise, accurately accounting for $N_2O$ emissions downstream of the
WWTPs is important for constraining and predicting future global $N_2O$ emissions. Efficient $N_2O$
removal, in addition to dissolved nitrogen removal, should be an essential part of water quality
control in WWTPs.

Key words: nitrous oxide, greenhouse gas emission, nitrogen pollution, wastewater treatment plants, spatial and seasonal variation


Summary: Wastewater treatment plants (WWTPs) are known to be hotspots of greenhouse gas emissions. However, the impact of WWTPs on the emission of the greenhouse gas $N_2O$ in downstream aquatic environments is less constrained. We found spatially and temporally variable but overall higher $N_2O$ concentrations and fluxes in waters downstream of WWTPs, pointing to the need for efficient $N_2O$ removal in addition to treating nitrogen in WWTPs.

41

 Graphical abstract

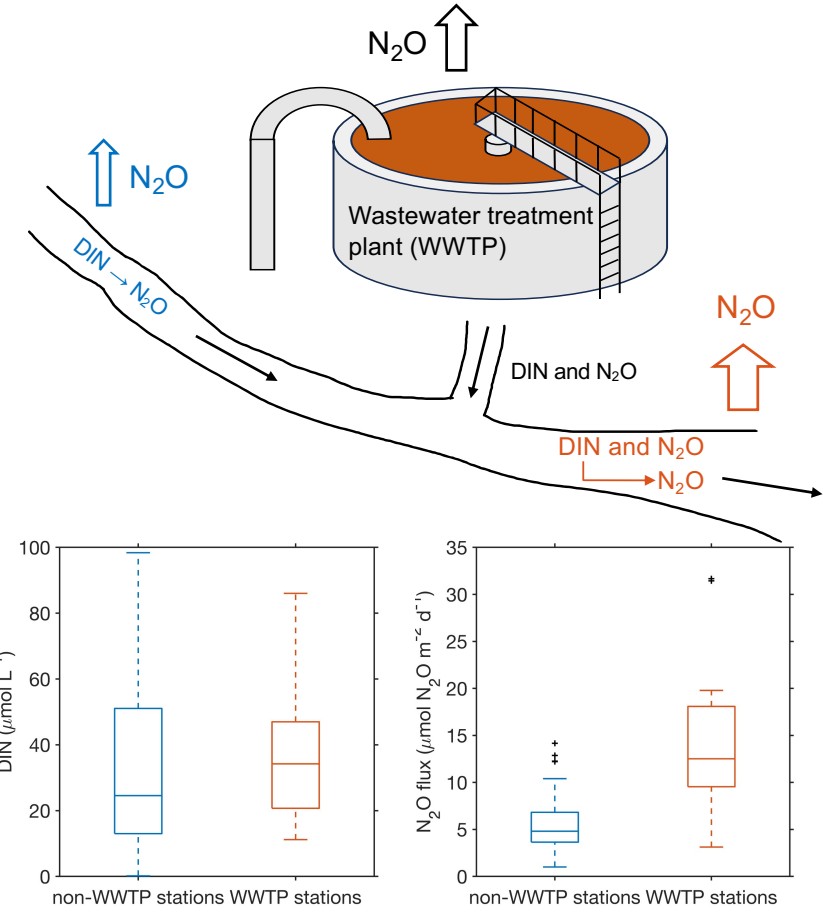

## Introduction

Nitrogen (N) enters the aquatic environment from agricultural and urban runoff, atmospheric deposition, and wastewater treatment plants (WWTPs), potentially leading to eutrophication, especially in densely populated regions (Galloway et al., 2008; Morée et al., 2013). During microbial transformations of N in aquatic systems (e.g., nitrification and denitrification), a powerful greenhouse gas and ozone depleting agent – $N_2O$ – is produced (Quick et al., 2019). Aquatic systems are large but highly variable sources of $N_2O$ to the atmosphere (Wang et al., 2023). For example, on a global basis, 0.04 - 0.291 Tg N $yr^{-1}$ and 0.04 - 3.6 Tg N $yr^{-1}$ of $N_2O$ is estimated to outgas from rivers and estuaries, respectively (Murray et al., 2015; Maavara et al., 2019; Yao et al., 2019; Rosentreter et al., 2023). The high end of the estimates in these inland and coastal waters approaches the scale of the global marine $N_2O$ emissions (2.5 - 4.3 Tg N $yr^{-1}$ in Tian et al., 2020). The large uncertainty in the estimate of aquatic $N_2O$ emission is partly due to high spatial and temporal variabilities of $N_2O$ flux within/across rivers and estuaries and the lack of observations to capture such variability. Therefore, sampling and measurements of $N_2O$ concentration at high spatial and temporal resolutions would be desirable to constrain aquatic $N_2O$ emission.

The major factors that appear to correlate with $N_2O$ concentration are dissolved inorganic nitrogen (DIN) and oxygen status (Hu et al., 2016; Zheng et al., 2022). Waste and wastewater release large amounts of DIN into the aquatic environment. In addition, waste and wastewater emit ~0.3 Tg N $yr^{-1}$ of $N_2O$ (estimated from 2007-2016) into the atmosphere globally, an amount that is continuously increasing at a rate of 0.04±0.01 Tg N $yr^{-1}$ per decade (Tian et al., 2020). $N_2O$ emission from WWTPs accounts for ~5.2% of total $N_2O$ emission in 2021 in the United States (EPA, 2023). $N_2O$ emissions from different WWTPs are highly variable, and are normally calculated as a function of DIN loading into the WWTPs, using an $N_2O$ emission factor (Kampschreur et al., 2009). $N_2O$ emission factors range from 0.16% to 4.5% ($N_2O$ emitted/DIN loading) (Eggleston et al., 2006; De Haas and Andrews, 2022). In addition to direct emission from the WWTPs, $N_2O$ can be discharged via WWTP effluent and produced due to DIN release from WWTP effluent into the creeks, rivers, and other downstream aquatic systems (McElroy et al., 1978; Beaulieu et al., 2010; Masuda et al., 2018). However, the impact of WWTPs on downstream $N_2O$ concentration is less studied and the downstream $N_2O$ emission remains poorly constrained.

Here we specifically compared the $N_2O$ concentration upstream and downstream of the WWTPs
in order to assess the impact of WWTPs on $N_2O$ emission, which could help to constrain the
emission factor associated with the WWTPs effluents.

The Potomac River is a major tributary of the Chesapeake Bay – the largest estuary in the United
States. The Potomac River Estuary is located in a highly populated area, mainly surrounded by
Washington, D.C., and the states of Virginia and Maryland in the eastern United States. The annual
mean discharge of Potomac River from 1895 to 2002 measured at Chain Bridge near Washington,
DC was 321 $m^3$ $s^{-1}$ with a large interannual variability (Jaworski et al., 2007). The annual total
nitrogen loading was estimated to be around 27.7 $\times 10^6$ kg N $year^{-1}$ in 2008-2009 (Bricker et al.,
2014). The Potomac River Estuary has experienced ecological degradation for decades partly due
to excess nutrient inputs including from the effluents of WWTPs (Bricker et al., 2014; Jaworski et
al., 2007). For example, the Blue Plains Advanced WWTP in Washington, D.C. is one of largest
WWTPs in the world, treating an average of ~1454 million liters of water per day. Pioneering
work in 1978 showed that Blue Plains WWTP was a large source of nitrogen to the Potomac River
Estuary, triggering high $N_2O$ production and concentration downstream (McElroy et al., 1978).
Thanks to higher standards mandated by governmental agencies (nitrogen concentration in
effluents below 7.5 mg $L^{-1}$) starting in 1980s and the technical improvements in N removal from
the wastewater, the nitrogen concentration in effluents of WWTPs in the Potomac River has
decreased substantially (Pennino et al., 2016). However, the concurrent effect on $N_2O$
concentration is largely unknown. The Department of Environmental Quality (DEQ) of Virginia
maintains an approximately monthly routine monitoring program for water quality (e.g., nitrogen
concentration, phosphorus concentration, chlorophyll concentration) and physical properties (e.g.,
temperature, salinity, pH, and dissolved oxygen concentration) in the Potomac River Estuary but
not for $N_2O$. Therefore, we collaborated with DEQ of Virginia to measure the spatial and temporal
variation of $N_2O$ concentrations in the Potomac River Estuary.

**Materials and Methods**
**Sample collection for $N_2O$ and nutrients**
Surface waters at ~0.5 m depth at eleven stations in the tidal Potomac River Estuary were sampled
monthly or bimonthly (depending on the weather) on a vessel (Grady White 208) for the analysis
of DIN concentration, and both concentration and nitrogen isotopes of $N_2O$ from April 2022 to
May 2023 (Figure 1). The eleven stations are characterized into 3 groups: embayment downstream
of WWTPs, embayment not associated with WWTPs, and the central channel of the Potomac
River. Three embayment stations downstream of WWTPs are associated with three different
WWTPs: Noman Cole, Mooney and Aquia, all of which implement tertiary treatment of the
wastewater. We obtained the volume discharge and total N in treated water of each WWTP from
Discharge Monitoring Reporting required by Virginia Pollutant Discharge Elimination System
permit. Noman Cole WWTP discharges ~140.8 million liters of water and 370 kg N per day into
Pohick Creek. Mooney WWTP discharges ~54.9 million liters of water and 147 kg N per day into
the Neabsco Creek. Aquia WWTP discharges much less water and N into the Aquia Creek (~21.2
million liters per day and 35 kg N per day). The distances from the embayment stations
downstream of WWTPs to Noman Cole, Mooney, Aquia WWTPs were approximately 4, 1.8 and
5.8 km, respectively.

The embayment stations were 2-3 meters deep while the average depth of central channel stations
was around 8 meters. The embayment stations have been routinely sampled for water quality
analyses by the DEQ of Virginia since the early 1970's. The central channel stations were added
for this study. The purposes of this sampling design are to evaluate the impact of WWTPs on
downstream distribution of DIN and $N_2O$, and to compare DIN and $N_2O$ concentrations between
edge and central channel of the river. The central channel is likely affected both by the Potomac
mainstem flow and by the input from tributaries, while the embayment stations may be mainly
affected by water flow from tributaries but also influenced by the tidal cycle (see the salinity
change in Supplementary Figure 1b). While estuarine $N_2O$ concentrations could be affected by
tides (Gonçalves et la., 2015), sampling was not always conducted at the same tidal state due to
logistic difficulties. Triplicate water samples for $N_2O$ concentrations and isotopes were collected
via a submersible pump into 60 mL serum bottles after overflowing three times the bottle's volume.
After removing 3 mL water to create a 3 mL air headspace via a syringe, the serum bottles were
immediately sealed with butyl stoppers and aluminum crimps and preserved with 0.5 mL of 10 M
NaOH solution to stop biological activities. NaOH has been shown to be an effective and less
environmentally hazardous preservative for $N_2O$ and nutrient analysis (Frame et al., 2016; Wong
et al., 2017).

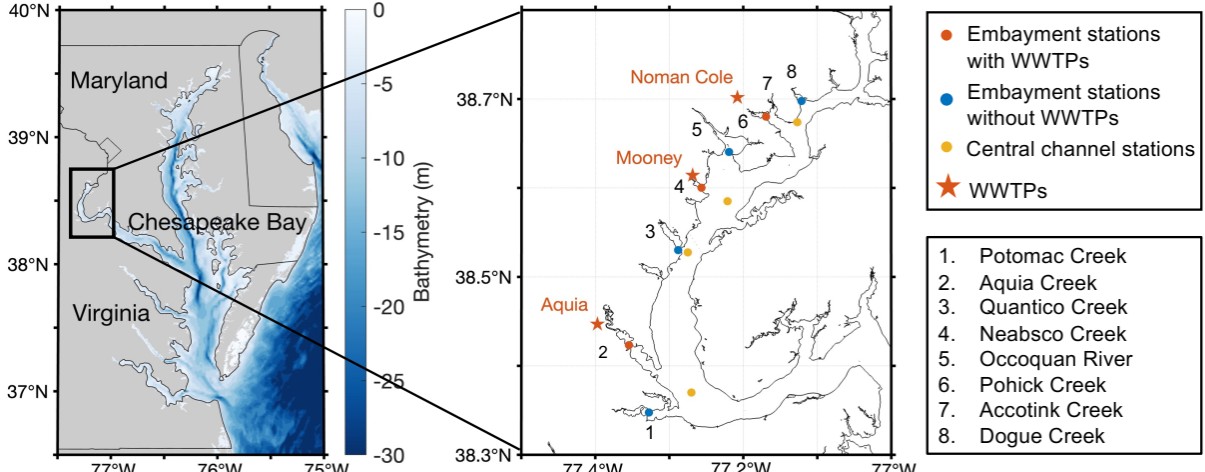


Figure 1. Sampling stations in the Potomac River Estuary including embayment stations associated with WWTPs (red circles) and without WWTPs (blue circles), and central channel stations (yellow circles). Locations of WWTPs (Noman Cole, Mooney and Aquia) are shown in red stars. Creeks/rivers with sampling stations are numbered in the map with names shown in the legend. Stream sampling sites upstream and downstream of WWTPs in creeks 4 – 7 are shown in Figure 4 below.

In addition to the routine sampling in the Potomac River Estuary, we also sampled its tributaries, some of which were associated with the WWTPs, on May 18, 2023 (Figure 1) to specifically evaluate the impact of WWTPs on downstream $N_2O$ concentrations. Four creeks/rivers were sampled including Neabsco Creek (5 stations: 2 stations upstream and 3 stations downstream of Mooney WWTP), Occoquan River (3 stations, no WWTP), Pohick Creek (4 stations: 2 stations upstream and 2 stations downstream of Noman Cole WWTP), and Accotink Creek (2 stations, no WWTP). Because Aquia WWTP discharges much less water and N into the Aquia Creek, its impact was not specifically investigated. Since water depths of these creeks/rivers were shallow, the water samples were collected by directly submerging 60 mL serum bottles into the surface water (~0.1 m) and preserving them as described above.

156

Besides $N_2O$ sampling, temperature, salinity, and dissolved $O_2$ concentrations were recorded via a YSI EXO1 sonde. Chlorophyll-a samples (300 mL) were filtered onto GF/F filters and kept on ice in a cooler. The filters were then kept frozen at -20℃ in the lab until analysis within 3 months (Arar and Collins, 1997). One additional sample for total nitrogen and phosphorus (both particulate and dissolved) was collected into 250 mL HDPE bottles and kept in ice in a cooler until analysis within 48 hours on land (Rice et al., 2012; EPA, 1983). Total nitrogen is the sum of total Kjeldahl nitrogen and nitrite plus nitrate.

**Measurement of $N_2O$ and nutrient concentrations**

$N_2O$ in the serum bottles was stripped by helium carrier gas into a Delta V Plus mass spectrometer (Thermo) for the analyses of $N_2O$ concentration and isotope ratio (m/z = 44, 45, 46) (Tang et al., 2022). The total amount of $N_2O$ in the serum bottles was determined using a standard curve of $N_2O$ peak area with $N_2O$ standards containing a known amount of $N_2O$ reference gas (0, 0.207, 0.415, 0.623, 0.831, 1.247 nmol $N_2O$). The total amount of $N_2O$ dissolved in the water was calculated after subtracting the amount of $N_2O$ in 3 mL air headspace. The amount of $N_2O$ in 3 mL air headspace was generally less than 4% of the amount of $N_2O$ dissolved in the 57 mL water samples. The $N_2O$ concentration in samples was then calculated from the total amount of $N_2O$ dissolved in the water divided by the volume of water in the serum bottles. The detection limit and precision of $N_2O$ concentration measurement were 1.29 and 0.33 nM, respectively. We used $N_2O$ produced from nitrate isotope standards (USGS34 = -1.8‰ and IAEA = 4.7‰) to calibrate for $\delta^{15}N$ of $N_2O$ samples. We then estimated $N_2O$ saturation (%): $\frac{N_2O_{measured}}{N_2O_{equilibrium}} \times 100$. The equilibrium $N_2O$ concentration ($N_2O_{equilibrium}$) was calculated based on the solubility of $N_2O$ and atmospheric $N_2O$ concentrations (Weiss and Price, 1980). The monthly atmospheric $N_2O$ concentrations were obtained from the nearby atmospheric station in Brentwood, Maryland (https://gml.noaa.gov/) (Andrews et al., 2023).

After analyzing $N_2O$ concentration, samples were neutralized to pH ~7 by adding 10% hydrochloric acid. $NO_2^- + NO_3^-$ ($NO_x^-$) concentration in these samples was measured using the vanadium (III) reduction method by converting $NO_x^-$ to NO, which was then quantified by chemiluminescence analyzer (Braman and Hendrix, 1989). The detection limit of $NO_x^-$

concentration was 0.15 µM. $NH_4^+$ and $NO_2^-$ concentrations were measured at a few selected
stations using the fluorometric orthophthalaldehyde method (Holmes et al., 1999) and the
colorimetric method (Hansen and Koroleff, 1999), respectively. Their concentrations were much
smaller than $NO_3^-$ alone, mostly accounting for less than 10% of the DIN concentration. Therefore,
we only present $NO_x^-$ data in this study.

**$N_2O$ flux calculation**
Surface $N_2O$ flux was calculated using the following equation: $Flux = k \times (N_2O_{measured} -$
$N_2O_{equilibrium})$. The gas transfer velocity ($k$) was estimated based on three different
parameterizations: $k = 1.91 \times e^{0.35 \times U} \times (\frac{Sc}{600})^{-0.5}$ (Raymond and Cole, 2001); $k = (0.314 \times$
$U^2 - 0.436 \times U + 3.99) \times (\frac{Sc}{600})^{-0.5}$ (Jiang et al., 2008); $k = 0.251 \times U^2 \times (\frac{Sc}{660})^{-0.5}$
(Wanninkhof, 2014). U is the wind speed at the 10 m height obtained from the National Centers
for Environmental Prediction (NCEP) reanalysis (Kalnay et al., 1996;
https://psl.noaa.gov/data/gridded/data.ncep.reanalysis.html). Sc is the Schmidt number that could
be estimated as a function of temperature (Wanninkhof, 2014). Since our samples have salinity
close to 0, we used the parameterization of Sc for freshwater. Average values of the three $N_2O$ flux
estimates are presented in the paper and $N_2O$ fluxes estimated by different parameterizations are
provided in the associated dataset. We acknowledge large variations in estimating $k$ values in the
riverine and estuarine systems by using different empirical models (Raymond and Cole, 2001;
Borges et al., 2004; Rosentreter et al., 2021). For instance, the effect of water velocity and water
depth on gas transfer velocity was not considered in the parameterizations above. Therefore, we
focus on evaluating the spatiotemporal variations in $N_2O$ fluxes and their driving factors instead
of their absolute magnitude.

**Results and discussion**
**Spatial and temporal variations of $N_2O$ concentrations in the Potomac River Estuary**
Along the roughly 50 km sampling transect in the Potomac River Estuary, $NO_x^-$ concentration
decreased from 98 to <1 µM from upstream to downstream (Figure 2a). $NO_x^-$ concentration
showed a clear seasonal pattern: higher in winter and spring while lower in summer and fall. The
spatial and temporal patterns were likely attributable to the distribution of nutrient sources into the
Potomac River, DIN uptake and other removal processes along the river (Glibert et al., 1995;
Carstensen et al., 2015). For example, the maximum N loading into the Chesapeake Bay occurs in
winter and spring (Da et al., 2018). Meanwhile, $N_2O$ concentration decreased from approximately
40 to 10 nM along the sampling transect and was higher in the fall and winter (Figure 2b). Since
temperature decreased from ~31°C in summer to 4°C in winter (Supplementary Figure 1a), the
increase in $N_2O$ solubility in colder water during winter partly explained the seasonal change. In
contrast, $N_2O$ saturation had higher values in summer and fall (Figure 2c), suggesting a higher
$N_2O$ production in summer and fall. It is worth noting that $N_2O$ saturation was above 100% at all
sampling stations with a maximum reaching 500%, indicating the Potomac River Estuary was a
consistent and strong source of $N_2O$ to the atmosphere. $N_2O$ fluxes ranged from 1 to 31.7 μmol
$N_2O$ $m^{-2}$ $d^{-1}$, generally decreasing from upstream to downstream (Figure 2d). $N_2O$ fluxes showed
a similar seasonal pattern to $N_2O$ saturation: higher in summer and fall. $N_2O$ concentrations
(median: 18.2 nM) and fluxes (median: 5.6 μmol $N_2O$ $m^{-2}$ $d^{-1}$) in the Potomac River Estuary were
substantially higher than in the mainstem of the Chesapeake Bay (2.6 to 20.9 nM $N_2O$ with a
median value at 10.6 nM and -0.3 to 4.3 μmol $N_2O$ $m^{-2}$ $d^{-1}$ with a median at 0.5 μmol $N_2O$ $m^{-2}$ $d^{-1}$
(Tang et al., 2022; Laperriere et al., 2019)). Therefore, the tributaries (i.e., Potomac River) are
more intense sources of $N_2O$ to the atmosphere than mainstem of the bay.

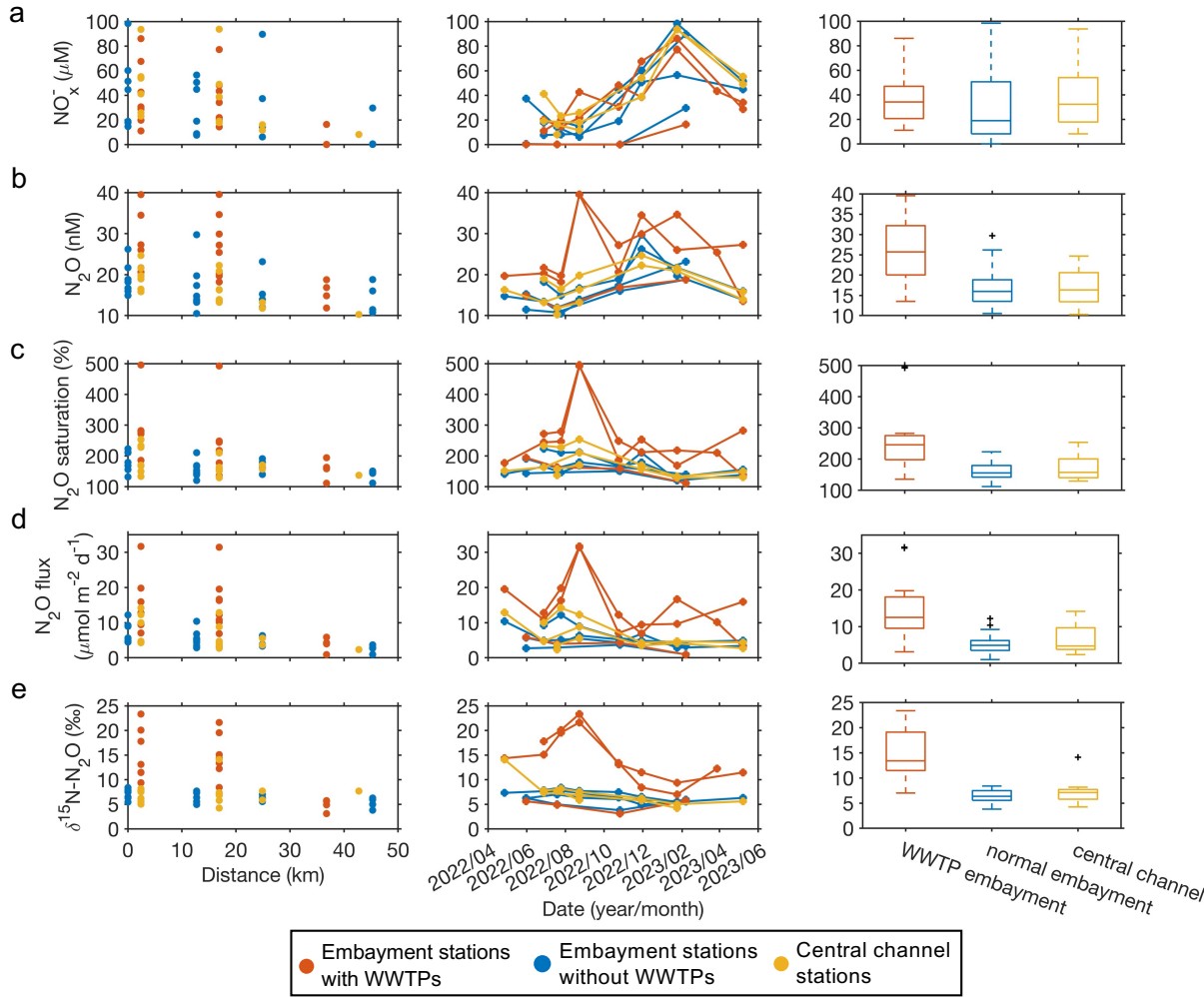

Figure 2. Spatial and temporal variations of $NO_x^-$ concentration (a), $N_2O$ concentration (b), $N_2O$ saturation (c), $N_2O$ flux (d) and $\delta^{15}N$ of $N_2O$ (e). The distance shows from upstream to downstream stations in the Potomac River. Embayment stations associated with WWTPs (red circles and lines) and without WWTPs (blue circles and lines), and central channel stations (yellow circles and lines). For the boxplots, the red line in each box is the median. The bottom and top of each box are the 25th and 75th percentiles of the observations, respectively. The error bars represent 1.5 times the interquartile range away from the bottom or top of the box, with black + signs showing outliers beyond that range. Embayment stations associated with WWTPs had significantly higher $N_2O$ concentration, $N_2O$ saturation, $N_2O$ flux and $\delta^{15}N$ values compared to other stations ($p<0.01$, $t$-test) but not significantly different $NO_x^-$ concentration.

Stations close to each other had similar $NO_x^-$ concentrations (e.g., upstream stations > downstream
stations), regardless of station category (i.e., with WWTP, without WWTP, central channel of the
Potomac River). In contrast, $N_2O$ concentrations and fluxes varied within locations according to
the station category: $N_2O$ concentrations and fluxes were substantially higher at stations
downstream of WWTPs (p<0.01, $t$-test). $N_2O$ concentrations and fluxes were similar between
stations in embayments without WWTPs and the central channel (Figure 2). This suggests these
WWTPs are efficient in removing DIN from sewage and other sources but WWTPs may discharge
$N_2O$ directly into the effluent or enhance downstream $N_2O$ production (e.g., higher $N_2O$ production
yield from the same amount of DIN). This effect extended to our sampling stations ~1.8-4 km
downstream of the WWTPs. However, the effect of WWTPs on downstream $N_2O$ varied among
stations. For example, elevated $N_2O$ concentrations were observed downstream from Noman Cole
and Mooney WWTPs but not downstream from Aquia WWTP. This difference may be related to
the different N removal processes of WWTPs that produce $N_2O$ at different yields (de Haas and
Andrews. 2022; Zhao et al., 2024). However, we don't have detailed information about the three
WWTPs other than that they all implement tertiary treatment. In addition, the different dilution
factors by riverine discharges also matter. For example, the volume of effluent from Mooney
WWTP was higher than the discharge of Neabsco Creek while the volume of effluent from Aquia
WWTP were generally lower than the discharge of Aquia Creek (Supplementary Figure 2a-b).
Particularly, the highest $N_2O$ concentration of up to 40 nM was found at two stations downstream
of the Noman Cole and Mooney WWTPs on August 23, 2022 when the river discharge was low
(Supplementary Figure 2). Thus, the effect of WWTPs on downstream $N_2O$ concentrations also
varies seasonally (Schult et al., 2023; Murray et al., 2020), with a relatively more important role
in the dry season. Repeated spatial and temporal sampling allowed us to capture these $N_2O$
hotspots. Previous studies have shown the impact of WWTPs on downstream $N_2O$ concentrations
and emissions in aquatic environments. For example, the highest $N_2O$ concentration ~675 nM in
the Potomac River was measured near the discharge of the Blue Plains WWTP in 1977 (McElroy
et al., 1978). Highest $N_2O$ emissions in the Ohio River near Cincinnati were attributed to direct
input of $N_2O$ from WWTPs' effluents (Beaulieu et al., 2010).

In addition, a higher nitrogen isotopic signature ($\delta^{15}N$) of $N_2O$ associated with WWTPs (median
$\delta^{15}N$ at 13‰) also suggests the distinct sources or cycling processes of $N_2O$ compared to stations
of the central channel and without the influence of WWTPs (median $\delta^{15}$N of N$_2$O at 6‰, Figure
2e) in the Potomac River Estuary. In comparison, the average $\delta^{15}$N of N$_2$O in the tropospheric air
is around 6.55‰ (Snider et al., 2015). $\delta^{15}$N of N$_2$O for stations with the influence of WWTPs
showed a clear seasonal variation: higher in summer than winter (Figure 2e). This seasonal
difference may be related to the seasonal change in the relative importance of WWTPs' effluents
versus riverine discharge (Supplementary Figure 2c). For example, a relatively larger WWTPs'
effluent volume compared to the riverine discharge led to a larger $\delta^{15}$N of N$_2$O in summer.
However, no clear seasonal pattern of $\delta^{15}$N of N$_2$O was seen for stations without the influence of
WWTPs. $\delta^{15}$N of N$_2$O produced in WWTPs depends on the treatment stages and aeration
conditions (Toyoda et al., 2011; Tumendelger et al., 2014). For example, the average $\delta^{15}$N values
were reported to be -24.5‰ and 0‰ respectively for N$_2$O produced from nitrification during oxic
treatment versus N$_2$O produced from anaerobic denitrification in a California WWTP (Townsend-
Small et al., 2011). The $\delta^{15}$N values of N$_2$O in these urban WWTPs were lower than those found
in waters downstream of WWTPs in the Potomac River (median $\delta^{15}$N at 13‰). One of the reasons
for the increased $\delta^{15}$N of N$_2$O may be partial N$_2$O reduction via denitrification in the WWTPs, in
downstream creeks, or in sediments; this denitrification effect has been seen in the marine oxygen
minimum zones (Kelly et al., 2021). Denitrification as the cause of the elevated $\delta^{15}$N is partly
supported by the higher $\delta^{15}$N of N$_2$O when NO$_x^-$ was reduced to less than 40 μM, suggesting the
occurrence of N$_2$O reduction when the concentration of other denitrification substrates became
low (Supplementary Figure 3). However, we do not know the exact locations where denitrification
occurred (e.g., WWTPs, anoxic niches in suspended particles, sediments), which deserves further
investigations. The influence of denitrification on unique isotopic signatures of N$_2$O produced
from WWTPs has also been observed in Tama River in Japan (Toyoda et al., 2009).

**Environmental controls on N$_2$O concentrations**
N$_2$O concentrations showed positive correlations with total N (r=0.62, p<0.01) and NO$_x^-$
concentrations (r=0.51, p<0.01) (Figure 3a). Correlation analyses done separately for stations with
or without WWTPs had similar patterns (Supplementary Figure 4). A better correlation between
the N$_2$O concentration and total N may indicate the contribution of other N sources besides NO$_x^-$
to N$_2$O production. N$_2$O could be produced from nitrification in the process of oxidizing NH$_4^+$ to
NO$_x^-$ in the oxic environment as previously shown in the oxygenated mainstem of the Chesapeake
Bay (Tang et al., 2022). However, we can't exclude the possibility of $N_2O$ production from
denitrification associated with anaerobic microsites in particles or in sediment (Beaulieu et al.,
2011; Wan et al., 2023). Future investigations with [15]N tracers should be conducted to differentiate
$N_2O$ production pathways around the WWTPs. Furthermore, $N_2O$ concentration was negatively
correlated with temperature since higher temperature reduced the $N_2O$ solubility. Although
previous studies have showed dissolved oxygen to be an important driver of $N_2O$ concentrations
or fluxes in rivers and estuaries (Rosamond et al., 2012; Wang et al., 2015; Zheng et al., 2022), we
did not find a strong dependence of $N_2O$ on oxygen concentrations in the Potomac River Estuary
(Figure 3a). This lack of strong dependence is probably because of the overall oxygenated
conditions (Supplementary Figure 1c), and opposite correlations found in stations without WWTPs
(positive) versus in stations with WWTPs (negative) (Supplementary Figure 4), which may be
influenced by the different $N_2O$ production pathways.

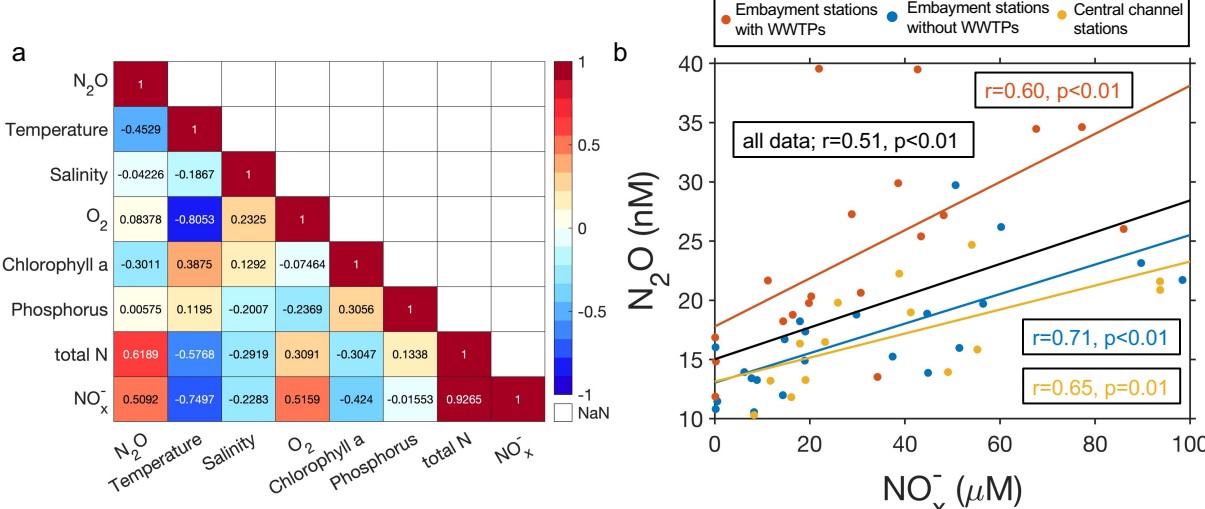


Figure 3. (a) Correlation coefficients among different environmental factors and $N_2O$
concentrations. (b) Relationship between $N_2O$ and $NO_x^-$ concentrations at different categories of
sampling stations.

The significant positive relationship between $N_2O$ and $NO_x^-$ concentration existed for samples
collected at stations from all three different categories (Figure 3b). $N_2O$ concentrations at stations
downstream of WWTPs were notably higher than at other stations not associated with WWTPs
even under the similar range of $NO_x^-$ concentration. The larger slope of $N_2O$ concentration versus
$NO_x^-$ concentration at stations downstream of WWTPs may be related to the direct input of $N_2O$
from WWTPs into the downstream waters or different $N_2O$ production pathways and production
yields that deserve further investigations. The DIN concentration has been found to be a good
predictor of $N_2O$ concentration and emission in many other rivers and estuaries (Murray et al.,
2015; Reading et al., 2020; Zheng et al., 2022;). However, the correlation varied spatially, which
may be affected by the variable $N_2O$ emission factors from DIN cycling. The emission factors are
affected by temperature, concentration and forms of N, oxygen, organic carbon concentration and
many other factors (Hu et al., 2016). The external $N_2O$ input (e.g., input from WWTPs) could also
affect the relationship between $N_2O$ and DIN concentrations (Dong et al., 2023). Compared to DIN
(~28 to 71 μM) and $N_2O$ concentrations (~16 to 61 nM) measured approximately 45 years ago in
the same section of the Potomac River (McElroy et al., 1978), current DIN and $N_2O$ concentrations
have slightly decreased. Thus, an additional benefit of nutrient regulation is the reduction of
greenhouse gas - $N_2O$ - emissions, beyond improving water quality.

Since $N_2O$ concentrations had the strongest correlation with total N concentrations (reflecting the
$N_2O$ production potential) and temperature (affecting $N_2O$ solubility), we developed a predictive
model of $N_2O$ concentration based on total N and temperature. Predictions were performed
separately for stations with WWTPs ($N_2O\ concentration = 0.115 \times total\ N - 0.241 \times$
$temperature + 17.185$, n=18, r=0.78; p<0.01) and without WWTPs including central channel
stations   ($N_2O\ concentration = 0.049 \times total\ N - 0.298 \times temperature + 18.888$,   n=23,
r=0.81, p<0.01). The observed $N_2O$ variability was generally captured by these simple linear
models (Supplementary Figure 5) but there were variabilities in the observations remaining to be
explained. Addition of other predictors did not significantly improve the model performance, so
we chose the simple predictive model that is mechanistically understandable. We then applied the
two predictive models separately to estimate $N_2O$ concentrations at the embayment station in the
Pohick Bay (with WWTP) and the embayment station in the Occoquan Bay (without WWTP)
using total N concentration and temperature that were measured since 2008 by the DEQ of Virginia
monitoring program (Supplementary Figures 6 and 7). Predicted $N_2O$ concentrations showed a
clear seasonality: higher in winter and lower in summer. $N_2O$ concentrations in the Pohick Bay
decreased substantially (-0.9 nM/year) possibly due to the nutrient reduction (total N concentration
decreasing at 8.8 μM/year) over the last 14 years (Supplementary Figure 6). However, $N_2O$
concentrations in the Occoquan Bay only decreased slightly (-0.1 nM /year, not statistically
significant) along with the minor nutrient reduction (total N concentration decreasing at non-
statistically significant rate of 0.5 μM/year) (Supplementary Figure 7). Continuation of
environmental monitoring in the Potomac River (e.g., N nutrients and temperature), which is much
easier than sampling and measuring $N_2O$ gas, could be used to indirectly estimate the changes in
$N_2O$ concentrations in the future. These predictors are likely to be important in other estuaries, but
the weighting would vary among locations.

**Impact of wastewater treatment plants on $N_2O$ concentrations and emissions**
To further evaluate how WWTPs affect the $N_2O$ distribution in the Potomac River, we measured
$N_2O$ concentrations upstream and downstream of the two WWTP effluents (Mooney and Noman
Cole in Neabsco Creek and Pohick Creek, respectively) and compared them to $N_2O$ concentrations
measured in two creeks that do not have WWTPs (Figure 4a). Interestingly, the $N_2O$ concentration
and flux at the station downstream of Mooney WWTP in Neabsco Creek were lower than the $N_2O$
concentration and flux at the station upstream of Mooney WWTP (15.0 nM vs 20.1 nM; 14.6 μmol
$m^{-2}$ $d^{-1}$ vs 24.7 μmol $m^{-2}$ $d^{-1}$). The exact mechanisms were not clear but one of the potential reasons
could be the influence by tidal cycles: high tide during the sampling time (salinity was 0.17 instead
of 0) may have reversed the water flow and diluted the WWTP effluent with low $N_2O$
concentration Potomac water (12.1 nM at the outflow of Neabsco Creek into the Potomac River
Estuary). In contrast, we found substantially a higher $N_2O$ concentration and flux downstream of
the Noman Cole WWTP than the upstream station (30.8 nM vs 16.7 nM; 55 μmol $m^{-2}$ $d^{-1}$ vs 17.6
μmol $m^{-2}$ $d^{-1}$) in the Pohick Creek, which is less affected by the tidal cycle due to its semi-closed
geography (salinity was 0.12). The high downstream $N_2O$ concentration and flux may suggest the
direct addition of $N_2O$ from WWTP effluent to the downstream environment. Furthermore, $\delta^{15}N$
of $N_2O$ in stations downstream of WWTPs were generally higher than the other two creeks that do
not have WWTPs (Figure 4b), confirming the distinct source of $N_2O$ production by WWTPs found
in the Potomac River Estuary. Overall, the influence of WWTP effluents on downstream
distribution of $N_2O$ is variable, and could be affected by the physical movement of water.

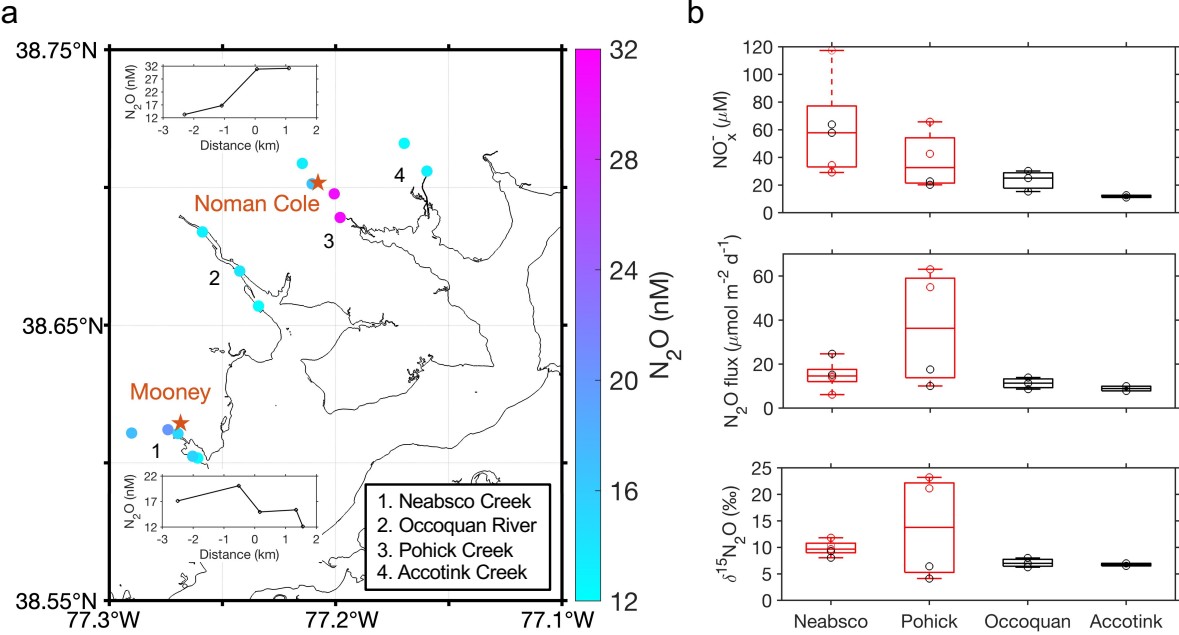

Figure 4. (a) Color-coded $N_2O$ concentration at creek sampling stations on May 18, 2023. WWTPs (Mooney and Noman Cole) are shown in red stars. The insert figures show the change in $N_2O$ concentrations as a function of distance up or down stream from the WWTPs. Creeks/rivers with sampling stations are numbered in the map with names shown in the legend. (b) Box plots of $NO_x^-$, $N_2O$ flux and $\delta^{15}N$ of $N_2O$ comparing four creeks. Neabsco and Pohick Creeks with WWTPs are displayed with red color boxes. Red and black circles in the boxplots show the data points of stations downstream and upstream/or without WWTPs, respectively. $NO_x^-$, $N_2O$ flux and $\delta^{15}N$ of $N_2O$ were clearly higher at stations downstream from the WWTP in Pohick Creek.

Dong et al. (2023) evaluated the potential impact of wastewater nitrogen discharge on estuarine $N_2O$ emissions globally. Here we compiled data from previous studies with direct $N_2O$ measurements in aquatic systems associated with WWTPs (not included in Dong et al., 2023) to assess the global impact of WWTPs on aquatic $N_2O$ concentrations or emissions (McElroy et al., 1978; Hemond and Duran, 1989; Toyoda et al., 2009; Beaulieu et al., 2010; Rosamond et al., 2012; Chun et al., 2020; Masuda et al., 2021; Masuda et al., 2018; Dylla, 2019). WWTP effluents or water downstream of the WWTPs contain some of the highest $N_2O$ concentrations and fluxes observed in the aquatic system (Table 1 and Supplementary Figure 8). For example, up to 12,411.4% saturation of $N_2O$ was measured in the effluent of WWTPs in the Tama River in Japan (Toyoda et al., 2009). In addition, $N_2O$ flux up to 40,800 $\mu mol$ $N_2O$-N $m^{-2}$ $d^{-1}$ was found

downstream of the Regina WWTP in the Wascana Creek in Canada (Dylla, 2019). The downstream $N_2O$ flux was >300 times higher than the $N_2O$ flux upstream of the Regina WWTP. In comparison, the maximum $N_2O$ saturation and flux previously reported in a global riverine $N_2O$ dataset were around 2,500% and 12,754 µmol $N_2O$-N $m^{-2}$ $d^{-1}$ (Hu et al., 2016). Across the sites listed in Table 1, $N_2O$ concentration/saturation/flux downstream of the WWTPs was 1.45 to 374-fold of the upstream waters. The only exception was our observed decrease in $N_2O$ concentrations downstream of Mooney WWTP on May 18, 2023, which was likely influenced by the tidal cycle. The wide range of apparent WWTP effect is related to many factors including the variable $N_2O$ emission factors in the WWTPs, the ratio of WWTP effluent volume to riverine discharge, the distance from the WWTPs where measurements were conducted, and the direction of water flow (e.g., tidal cycle). In addition, the estuarine type, mixing regime, and stratification are also important factors controlling $N_2O$ emissions (Brown et al., 2022). Overall, failing to account for $N_2O$ emissions downstream of the WWTPs and their variability would substantially bias estimates of aquatic $N_2O$ emissions. This uncertainty is increased by the fact that only a few observations are available (all in the northern hemisphere) (Supplementary Figure 8) compared to >58 000 WWTPs present globally (Ehalt Macedo et al., 2022). It is also important to restrict the $N_2O$ emission via efficient $N_2O$ reduction in the WWTPs considering the projected increase in future wastewater production (Qadir et al., 2020).

Table 1. Global $N_2O$ observations in aquatic systems associated with wastewater treatment plants. $N_2O$ data are presented in concentration (nM), saturation (%) or flux (µmol $N_2O$-N $m^{-2}$ $d^{-1}$) according to how they are reported in different studies. *downstream vs upstream.

| River/location | WWTP | $N_2O$ upstream or in tributaries without WWTPs | $N_2O$ in WWTP effluents | $N_2O$ downstream or in tributaries with WWTPs | *Average fold change | Reference |
|---|---|---|---|---|---|---|
| Potomac River/ Washington, D.C., USA | Blue Plains WWTP | 11-34 nM | | 147-318 nM | 9.3 | McElroy et al., 1978 |
| Assabet River/ Massachusetts, USA | Westborough WWTP | ~10 nM | 1045 nM | 163 nM | 16.3 | Hemond and Duran. 1989 |
| Tama River/ Tokyo, Japan | Plant 1 Plant 2 | 350.7% 219.3% | 12411.4% 3326.2% | 3454.8% 1029.6% | 9.8 4.7 | Toyoda et al., 2009 |
| Ohio River/ Cincinnati, USA | | 27.9 µmol $N_2O$-N $m^{-2}$ $d^{-1}$ | | 1068 µmol $N_2O$-N $m^{-2}$ $d^{-1}$ | 38.2 | Beaulieu et al., 2010 |

| | | | | | | |
|---|---|---|---|---|---|---|
| Grand River/ Ontario, Canada | e.g., Kitchener WWTP | 4-12 $\mu$mol $N_2O$-N $m^{-2}$ $d^{-1}$ | | 9-113 $\mu$mol $N_2O$-N $m^{-2}$ $d^{-1}$ | 9.4 | Rosamond et al., 2012 |
| Wascana Creek/ Saskatchewan, Canada | Regina WWTP | –32.5 to 109 $\mu$mol $N_2O$-N $m^{-2}$ $d^{-1}$ | 227 to 72800 $\mu$mol $N_2O$-N $m^{-2}$ $d^{-1}$ | 398 to 40800 $\mu$mol $N_2O$-N $m^{-2}$ $d^{-1}$ | 374 | Dylla. 2019 |
| Han River/ Seoul, Korea | JNW | 39.7 nM | 602.1 nM | 441.6 nM | 11.1 | Chun et al., 2020 |
| A-river B-river C-river/Miyagi, Japan | A-WWPT B-WWTP C-WWTP | 61 nM 95 nM 100 nM | 493 nM 246 nM 319 nM | 180 nM 286 nM 145 nM | 3 3 1.45 | Masuda et al., 2021 Masuda et al., 2018 |
| Potomac River Estuary /Virginia, USA | Noman Cole Mooney Aqua | 10.8-29.7 nM 1-12.2 $\mu$mol $N_2O$-N $m^{-2}$ $d^{-1}$ | | 11.87-39.5 nM 0.95-31.7 $\mu$mol $N_2O$-N $m^{-2}$ $d^{-1}$ | 1.6 2.2 | This study |
| Neabsco Creek/ Virginia, USA | Mooney | 20.1 nM 24.7 $\mu$mol $N_2O$-N $m^{-2}$ $d^{-1}$ | | 15.0 nM 14.6 $\mu$mol $N_2O$-N $m^{-2}$ $d^{-1}$ | 0.75 0.59 | This study |
| Pohick Creek/ Virginia, USA | Noman Cole | 16.7 nM 17.6 $\mu$mol $N_2O$-N $m^{-2}$ $d^{-1}$ | | 30.8 nM 55 $\mu$mol $N_2O$-N $m^{-2}$ $d^{-1}$ | 1.84 3.12 | This study |

## Conclusion

Taking advantage of the routine water monitoring program by the DEQ of Virginia, we detected strong spatial and temporal variabilities of $N_2O$ concentrations and emissions in the Potomac River Estuary, a major tributary of Chesapeake Bay. Observations across the Potomac River Estuary also allowed us to identify hotspots of $N_2O$ emissions associated with WWTPs effluents. Higher $N_2O$ concentrations downstream of WWTPs compared to regions with similar nitrogen nutrient concentrations suggested the direct discharge of dissolved $N_2O$ from WWTPs and/or intense $N_2O$ production. The influence of WWTPs on downstream $N_2O$ concentrations and emissions is largely affected by volumes of river discharges versus WWTPs effluents. A survey of globally available data shows $N_2O$ concentrations or emissions are consistently elevated in waters downstream from WWTPs. Future [15]N tracer incubations would help to explain the high $N_2O$ concentration downstream of WWTPs by disentangling the $N_2O$ production pathways. In addition, concurrent measurements of the N flux and $N_2O$ concentration downstream of WWTPs will help to constrain overall $N_2O$ emission factors associated with WWTPs. Our work could encourage potential collaborations between scientific community and governmental agencies/the public to better observe the environmental pollution or quality, e.g., increasing the frequency and resolution of observations for $N_2O$ and other greenhouse gases along with many regularly monitored

environmental factors like temperature and nutrients. Such efforts may identify previously overlooked sources of $N_2O$ emission and help to better estimate $N_2O$ emissions from aquatic systems.

## Data availability

Data presented in this study has been deposited in Zenodo repository: https://doi.org/10.5281/zenodo.10775250.

## Author contribution

W.T. conceived the study. J.T., T.J., and W.T. collected $N_2O$ samples from the Potomac River Estuary. W.T. analyzed samples and interpreted data with other coauthors. W.T. wrote the first draft of the manuscript with input from B.B.W. All coauthors contributed to the result discussion and manuscript writing.

## Competing interests

The authors declare that they have no conflict of interest.

## Acknowledgements

We thank Catherine Hexter for the help with water sampling in the tributaries of Potomac River on May 18, 2023. We thank Elizabeth Wallace and Lindsay Pagaduan for analyzing the nutrient samples. We thank Virginia Department of Environmental Quality for maintaining the routine sampling and for providing the opportunity to collect $N_2O$ samples in the Potomac River Estuary. We thank Virginia Pollutant Discharge Elimination System for providing water discharge and quality data of wastewater treatment plants. This study is supported by Princeton University.

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
