# Peer review of "Variable contribution of wastewater treatment plant effluents to downstream nitrous"

_EGUsphere, 2024_

## Referee Comment (RC1)

**GENERAL COMMENTS**

Currently, there is considerable interest in understanding the production and emissions of greenhouse gases (GHGs) from wastewater treatment plants (WWTPs). Indeed, emissions of these gases have become a major concern in efforts to mitigate climate change and reduce global emissions. This renders the article by Tang et al. particularly significant as it investigates the impact of WWTPs on $N_2O$ emissions in aquatic systems downstream of the Potomac River estuary by measuring nitrogenous nutrients and $N_2O$ concentrations on a monthly resolution over the course of a year. The authors have identified spatially and temporally variable concentrations of $N_2O$ and fluxes of $N_2O$, generally higher downstream of the WWTPs, highlighting the necessity for effective $N_2O$ removal alongside nitrogen treatment at WWTPs.

The data are well presented and the discussion of the dataset is comprehensive and conclusive. However, from my point of view, I have some suggestions to render the work more attractive to readers. Therefore, I suggest its publication after major revisions.

It would be valuable for the study to clarify whether the three treatment plants (Noman Cole, Mooney, and Aquia) utilize identical wastewater treatment processes and treat similar volumes of water. Additionally, assessing whether the receiving channels into which the WWTPs discharge exhibit comparable water volumes is crucial for ensuring a consistent dilution effect of the gas in the water column. Moreover, understanding the depth of the water column is essential; in cases of shallow depths, the influence of gas emission from the sediment to the water column could be substantial.

It would be interesting for the study to elucidate whether the three treatment plants (Noman Cole, Mooney, and Aquia) employ the same wastewater treatment processes and the volume of water they treat. It is also important to determine if the receiving channels where the WWTPs discharge have similar water volumes, so that the dilution effect of the gas in the water column is similar. Similarly, it would be interesting to know the depth of the water column; if it is shallow, the influence of gas emission from the sediment to the water column could be significant.

The bibliographical references cited do not always follow the same criteria (chronological order or alphabetical order).

**SPECIFIC COMMENTS**

Ln 49. In a more recent article than those cited, Rosentreter et al., 2023 there are compiled $N_2O$ emissions data from various estuaries, providing a wider range of emissions variation (0.2 – 5.7 Tg $N_2O$ yr$^{-1}$). Specifically, the paper states: "*Global estimates of estuarine $N_2O$ emissions are highly uncertain, with large discrepancies for both observation-based (220–5,710 $GgN_2O\,yr^{-1}$) and modelling approaches (94–1,084 $GgN_2O\,yr^{-1}$).*

Ln 65-66. References should be listed in ascending chronological order, consistent with the rest of the paper.

Ln 87. It should be indicated what type of treatment is given in the WWTPs (primary, secondary, tertiary, etc.) in order to understand if the nitrogen removal capacity of the three wastewater treatment plants is the same. At what distance from the WWTPs were the samples taken? Were the samples taken at approximately the same distance from the discharge point at all three WWTPs? Were the channels where the samples were taken similar? Did they have approximately the same water volume? An important factor when comparing the amount of $N_2O$ in the receiving channels is dilution.

Ln 99. Were the samples collected from a vessel? Please specify.

Ln 110-11. How was a 3 mL air headspace created in the 60 mL serum bottles? Did all samples have exactly the same volume of air headspace? Could this 3 mL of air in contact with the sample potentially interfere with the measurement? Was the $N_2O$ content in the air also measured? Were the samples taken in duplicate?

Ln 112. Leave a space between the 10 and the M.

Ln 124-128. Figure 4a, depicting the sampling points of the four streams/rivers (Neabsco Creek (5 stations), Occoquan River (3 stations), Pohick Creek (4 stations), and Accotink Creek), should be included in the Materials and Methods section.

Ln 128. Where have the data on water discharge and nitrogen (kg) per day from the wastewater treatment plants been obtained? It would be interesting to include this information in the manuscript.

Ln 149. It is not reflected in the text how the 3 mL air headspace is taken from the serum bottles to estimate the amount of $N_2O$ in the sample.

Ln 167. *"The equilibrium $N_2O$ concentration was calculated based on the solubility of N2O (Weiss and Price, 1980)..."* Where did you obtain the value of $N_2O$ in the atmosphere for the calculations? Which value did you consider, the daily, monthly...?

Ln 170: What do the initials NCEP stand for? It would be more comprehensive to include the website from which the value of U was taken.

Ln 171. You should cite in the paper the expression from which Sc has been estimated, possibly from the proposed expression by Wanninkhof (2014). You should indicate whether for the calculation of Sc, you have considered the expression for salinity equal to zero, or if, on the contrary, the $N_2O$ Schmidt number for each point has been scaled to the values proposed by Wanninkhof (2014) for salinities between 0 and 35, assuming that Sc varies linearly with salinity.

Ln 173. References should be listed in ascending chronological order, consistent with the rest of the paper.

Ln 170-176. I don't understand why they are using a gas transfer velocity parameterization (k) proposed for the ocean, such as the expression by Wanninkhof (2014), rather than a k for a coastal system. If they didn't have data on current velocity and depth of the system necessary to use the k by Borges et al. (2004) and Rosentreter et al. (2021), they could have used the expression proposed by Raymond and Cole (2001), which is based on a compilation of k proposed for different coastal systems, or that by Jiang et al. (2008), based on the compilation of Raymond and Cole (2001) as well as other studies conducted in estuaries. Furthermore, given the uncertainty associated with k, to minimize this, they could have estimated water-atmosphere fluxes considering two expressions of k (Raymond and Cole, 2001; Jiang et al., 2008; Wanninkhof, 2014), and taken the average value of the three fluxes obtained, as many other authors do in coastal systems (e.g., Call et al., 2015; Sánchez-Rodriguez et al., 2024)

Ln 233. I suggest wording it like this: ….vs 6‰ for stations of the central channel and without the influence by WWTPs

Ln 242. In general, denitrification typically occurs in environments with low oxygen concentrations (DO $\leq$ 5 μmol L−1, Codispoti et al., 2001). As illustrated in Supplementary Figure 2, oxygen concentrations at the stations never reached low values. In fact, downstream stations of wastewater treatment plants exhibited dissolved oxygen levels ranging between 139.38 (25/07/2022) – 430.94 μM (7/02/2023). It is recognized that denitrification can also take place within oxygenated water columns containing suspended organic matter particles (Bange, 2008). Is there a substantial amount of suspended material in the studied system that could induce denitrification in oxygenated water? On the other hand, it is well-established that coastal sediments provide optimal environments for denitrification due to continuous inputs of nutrients and organic matter from land. Could it be that some of the measured $N_2O$ in the water originates from the sediment?

Ln 252. Correlations of 0.62 ($r^2$=0.38) and 0.51 ($r^2$=0.26), I do not consider them strong correlations, remove the word strong.

Ln 252-256 and 263-264. In stations unaffected by WWTPs, there appears to be a good positive correlation between $N_2O$ and DO and NOx, which could indicate that nitrification is an important process in these $N_2O$ production stations.

Ln 262. References should be listed in ascending chronological order, consistent with the rest of the paper.

Ln 278-279. References should be listed in ascending chronological order, consistent with the rest of the paper.

Ln 292-294. Why does it not also present the predictive model of $N_2O$ concentration based on total nitrogen and temperature for stations in the central channel of the Potomac Estuary? It could be interesting to have it to extrapolate to other areas of the estuary located in the channel. Perhaps you have included the data measured in the channel in the

samples without wastewater treatment plants (WWTPs). If so, please indicate it. I believe you should have stated the number of stations/data considered in each prediction.

Ln 298-300. Did you use the prediction model for stations without WWTPs? Please indicate it in the paper.

**FIGURES**

Figure 2 and Supplementary Figure 1. What does the "01" after the slash indicate on the x-axis of the central graphs? Wouldn't it be more intuitive for the reader to use "22" or "23" instead of "01," depending on the year the sampling was conducted?

Supplementary Figure 2. In the figure caption and in Figure a, a negative sign as a subscript is missing on $NOx^-$ on the x-axis. In Figure b, on the x-axis, remove the space between $N_2$ and O.

**REFERENCES:**

Bange, H. W., 2008. Gaseous Nitrogen Compounds ($NO$, $N_2O$, $N_2$, $NH_3$) in the Ocean. In Nitrogen in the Marine Environment (pp. 51–94). Elsevier Inc. https://doi.org/10.1016/B978-0-12-372522-6.00002-5

Call, M., Maher, D. T., Santos, I. R., Ruiz-Halpern, S., Mangion, P., Sanders, C. J., ... & Eyre, B. D., 2015. Spatial and temporal variability of carbon dioxide and methane fluxes over semi-diurnal and spring–neap–spring timescales in a mangrove creek. Geochimica et Cosmochimica Acta, 150, 211-225.

Codispoti, L.A., Brandes, J.A., Christensen, J.P., Devol, A.H., Naqvi, S.W.A., Paerl, H.W., Yoshinari, T., 2001. The oceanic fixed nitrogen and nitrous oxide budgets: moving targets as we enter the anthropocene? Sci. Mar. 65 (S2), 85–105.

Jiang, L.Q., Cai, W.J., & Wang, Y. (2008). A comparative study of carbon dioxide degassing in river- and marine-dominated estuaries. Limnology and Oceanography, 53(6), 2603–2615.

Rosentreter, J. A., Laruelle, G. G., Bange, H. W., Bianchi, T. S., Busecke, J. J. M., Cai, W. J., Eyre, B. D., Forbrich, I., Kwon, E. Y., Maavara, T., Moosdorf, N., Najjar, R. G., Sarma, V. V. S. S., Van Dam, B., & Regnier, P., 2023. Coastal vegetation and estuaries are collectively a greenhouse gas sink. Nature Climate Change, 13(6), 579–587. https://doi.org/10.1038/s41558-023-01682-9

Sánchez-Rodríguez, J., Ortega, T., Sierra, A., Mestre, M., Ponce, R., Fernández-Puga, M. D. C., & Forja, J., 2024. Distribution, reactivity and vertical fluxes of methane in the Guadalquivir Estuary (SW Spain). Science of The Total Environment, 907, 167758.

---

## Referee Comment (RC2)

The manuscript "Variable contribution of wastewater treatment plan effluents to nitrous oxide emission" by Tang et al. studies the effects of wastewater treatment plants on the Potomac River estuary in the United States. For over one year, they took monthly samples for nitrous oxide, total nitrogen and dissolved inorganic nitrogen concentrations. Generally, the results showed spatial and seasonal variability in nitrous oxide concentrations with higher concentrations downstream of the WWTPs, highlighting the importance of WWTPs regarding estuarine N$_2$O emissions. Therefore, this manuscript will be of interest in the context of global N$_2$O emissions from estuaries and WWTPs.

The data set is well presented and interpreted and the text well written and organized. However, major revisions are necessary to discuss effects of wastewater treatment processes and dilution effects.

**General remarks:**

The paper misses to discuss differences in wastewater treatments and dilution effects, which leads to some important unanswered questions:

- Do the WTTPs differ in type, removal strategy and treated water volume? Are differences visible in TN, DIN and N$_2$O effluents?
- How big are the water volumes of the WTTP effluents compared to the water volume in the estuary (especially in the tributaries)? I would recommend calculating a wastewater discharge fraction of stream flow.
- How big is the N load in the WTTP effluents compared to the N loads in the upstream river? How are the effluents diluted and are concentration increases expected/seen?
- Are there seasonal effects on the impact of wastewater effluents? For example, Murray et al. (2020) measured differences in N$_2$O concentrations affected by WWTPS between dry and wet season in an Australian estuary.

**Specific remarks:**

L63: "[…] are highly variable, and are normally […]"

L75: What is the mean annual discharge entering the estuary from the upstream river? What are mean N loads?

L84: "[…] nitrogen effluent concentration below 7.5 mg L$^{-1}$ […]"

L108: At what tidal state was the sampling carried out? How does the tidal state affect the results? Did you always sampled at the same tidal state to minimize effects?

L110: Did you take replicates?

L110-111: Did you measure N$_2$O concentrations in air headspace for correction? How did you estimate/measure atmospheric N$_2$O concentrations?

L151: Did you measure replicates for N$_2$O isotopes?

L169-170: Why did you decide to use Wannikhof's formula, which applies better to open oceans? There are formulas specifically designed for estuarine environments, e.g. Clark et al. (1995) and Raymond and Cole (2001).

L128-131: How do these values (treated water volumes and N loads) compared to the riverine volume and N loads? See general comments above. Did you see changing impacts depending on the size of the WWTPs?

L149: How did you take the amount of N$_2$O in the 3 mL headspace into account?

L171: How did you calculate the Schmidt number?

L185: Do you also see these seasonal differences in the effect of the WWTPs? The effluent of WWTPs usually have a relatively constant N load throughout the entire year. Therefore, I could imagine that it makes a big difference whether the WWTPs discharge into an estuary with a high N concentration in winter or a low N concentration in summer. Further, riverine discharge is usually higher in winter, which leads to greater dilution and reduces the impact of WWTP effluents.

L190-191: Does this also reflects in seasonal changing $\delta^{15}N$-$N_2O$ values?

L218: Calculating a wastewater discharge fraction of stream flow would help to estimate the different dilution effects for each WWTP.

L220: Can you estimate the wastewater discharge fraction of stream flow considering the water volume of the estuary and water volume and N load from the WWTP?

L224: "High-resolution spatial and temporal sampling" – I don't agree that the conducted sampling campaign has a high spatial and temporal resolution considering the existence of laser-based measurements that allow resolution by the second. Sampling was conducted once or twice a month at eleven stations or once at 14 stations. I would suggest rephrasing this statement.

L233: Do you observe seasonal changes?

L238: What kind of treatments are performed at the WWTPs discharging into the Potomac River estuary? There are different ways of operating N removal within WWTP (biological, chemical, and physical methods) (e.g. Winkler and Straka, 2019; Zhou et al., 2023). Further, biological removal strategies, for example, can also differ significantly: (1) denitrification followed by nitrification, where a part of the treated water is fed back into the denitrification after nitrification, (2) nitrification is followed by denitrification with organic carbon being added to the denitrification chamber (e.g. part of the untreated water before nitrification), (3) intermittent denitrification, in which longer phases with aerobic nitrification and anoxic denitrification alternate in the same tank, (4) simultaneous denitrification due to the discontinuous or punctual supply of oxygen, (5) cascade denitrification, in which the wastewater passes through several tanks with alternating denitrification and nitrification, or (6) alternating denitrification, consisting of two aeration tanks that are alternately fed with wastewater and aerated. $N_2O$ production and $N_2O$ production pathway may differ significantly depending on the treatment strategy. Therefore, it would be very valuable to discuss treatment strategies considering possible isotope changes. Do the WWTP even use biological treatments or other physical/chemical ones?

L242: Oxygen concentration during your measurements (supplementary material Fig. 1, L264) were always above the threshold for denitrification (< 6.25 µM; Seitzinger, 1988). Denitrification can occur in anoxic microsites close to particles (Liu et al., 2013; Zhu et al., 2018; Schulz et al., 2022) or in anoxic sediments. Where do you suggest denitrification occurs? Is it an artefact of denitrification in the WWTP?

L250: Not a strong (r = 0.51), but a significant correlation (p<0.01) – Thus, I would rephrase "$N_2O$ concentrations showed a significant positive correlation [...]"

L254: Did you observe correlations between $NH_4^+$ and/or $NO_2^-$ concentrations with $N_2O$?

Figure 3: Why is Chlorophyll a in brackets?

L292: "WWTPs"

L299: Did you use the prediction with or without WWTPs?

L317: Did you consider tidal state during your sampling (e.g. always sampled at same tidal state)?

L334-335: Remove space between "$NO_x^-$" and ","

L357-359: Brown et al. (2022) also found estuarine type, mixing regime and stratification important factors controlling $N_2O$ emissions.

Supplementary Material S24: "$\delta^{15}N$ of $NO_x$ concentration (a) and $N_2O$ concentration (b)"

Supplementary Material Fig. 3: Why is Chlorophyll a in brackets?

Supplementary Material L33: "[…] the influence of WWTPs […]"

**References**

Brown, A. M., Bass, A. M., and Pickard, A. E.: Anthropogenic-estuarine interactions cause disproportionate greenhouse gas production: A review of the evidence base, Mar. Pollut. Bull., 174, 113240, https://doi.org/10.1016/j.marpolbul.2021.113240, 2022.

Clark, J. F., Schlosser, P., Simpson, H. J., Stute, M., Wanninkhof, R., and Ho, D. T.: Relationship between gas transfer velocities and wind speeds in the tidal Hudson River determined by the dual tracer technique, in: Air-Water Gas Transfer, edited by: Jähne, B. and Monahan, E. C., AEON Verlag, Hanau, 785–800, 1995.

Liu, T., Xia, X., Liu, S., Mou, X., and Qiu, Y.: Acceleration of denitrification in turbid rivers due to denitrification occurring on suspended sediment in oxic waters, Environ. Sci. Technol., 47, 4053–4061, https://doi.org/10.1021/es304504m, 2013.

Murray, R. H., Erler, D. V., Rosentreter, J., Wells, N. S., and Eyre, B. D.: Seasonal and spatial controls on N2O concentrations and emissions in low-nitrogen estuaries: Evidence from three tropical systems, Mar. Chem., 221, 103779, https://doi.org/10.1016/j.marchem.2020.103779, 2020.

Raymond, P. A. and Cole, J. J.: Gas exchange in rivers and estuaries: Choosing a gas transfer velocity, Estuaries, 24, 312–317, https://doi.org/10.2307/1352954, 2001.

Schulz, G., Sanders, T., van Beusekom, J. E. E., Voynova, Y. G., Schöl, A., and Dähnke, K.: Suspended particulate matter drives the spatial segregation of nitrogen turnover along the hyper-turbid Ems estuary, Biogeosciences, 19, 2007–2024, https://doi.org/10.5194/bg-19-2007-2022, 2022.

Seitzinger, S. P.: Denitrification in freshwater and coastal marine ecosystems: Ecological and geochemical significance, Limnol. Oceanogr., 33, 702–724, https://doi.org/10.4319/lo.1988.33.4part2.0702, 1988.

Winkler, M. K. H. and Straka, L.: New directions in biological nitrogen removal and recovery from wastewater, Curr. Opin. Biotechnol., 57, 50–55, https://doi.org/10.1016/j.copbio.2018.12.007, 2019.

Zhou, Y., Zhu, Y., Aber, J., Li, C., and Chen, G.: A Comprehensive Review on Wastewater Nitrogen Removal and Its Recovery Processes, Int. J. Environ. Res. Public. Health, 20, 3429, https://doi.org/10.3390/ijerph20043429, 2023.

Zhu, W., Wang, C., Hill, J., He, Y., Tao, B., Mao, Z., and Wu, W.: A missing link in the estuarine nitrogen cycle?: Coupled nitrification-denitrification mediated by suspended particulate matter, Sci. Rep., 8, 2282, https://doi.org/10.1038/s41598-018-20688-4, 2018.

---

## Author Comment (AC1)

Reviewer 1:

GENERAL COMMENTS

Currently, there is considerable interest in understanding the production and emissions of greenhouse gases (GHGs) from wastewater treatment plants (WWTPs). Indeed, emissions of these gases have become a major concern in efforts to mitigate climate change and reduce global emissions. This renders the article by Tang et al. particularly significant as it investigates the impact of WWTPs on N2O emissions in aquatic systems downstream of the Potomac River estuary by measuring nitrogenous nutrients and N2O concentrations on a monthly resolution over the course of a year. The authors have identified spatially and temporally variable concentrations of N2O and fluxes of N2O, generally higher downstream of the WWTPs, highlighting the necessity for effective N2O removal alongside nitrogen treatment at WWTPs.

The data are well presented and the discussion of the dataset is comprehensive and conclusive. However, from my point of view, I have some suggestions to render the work more attractive to readers. Therefore, I suggest its publication after major revisions.

It would be valuable for the study to clarify whether the three treatment plants (Noman Cole, Mooney, and Aquia) utilize identical wastewater treatment processes and treat similar volumes of water. Additionally, assessing whether the receiving channels into which the WWTPs discharge exhibit comparable water volumes is crucial for ensuring a consistent dilution effect of the gas in the water column. Moreover, understanding the depth of the water column is essential; in cases of shallow depths, the influence of gas emission from the sediment to the water column could be substantial.

It would be interesting for the study to elucidate whether the three treatment plants (Noman Cole, Mooney, and Aquia) employ the same wastewater treatment processes and the volume of water they treat. It is also important to determine if the receiving channels where the WWTPs discharge have similar water volumes, so that the dilution effect of the gas in the water column is similar. Similarly, it would be interesting to know the depth of the water column; if it is shallow, the influence of gas emission from the sediment to the water column could be significant.

We thank the reviewer for their valuable and insightful comments! The main suggestions include the quantification of the dilution of WWTPs effluents by river flows and better estimates of N$_2$O emissions using multiple gas transfer coefficient parameterizations. We have responded to reviewers' comments below in blue font and made changes accordingly in the manuscript.

Although we contacted the WWTPs directly, we were not able to obtain detailed information about the treatment processes of the three treatment plants except they all implement tertiary treatment. We acknowledged that the different types of treatment affect the N$_2$O production yield in the WWTPs in the text (de Haas and Andrews. 2022; Zhao et al., 2024).

For evaluating the dilution effect, we obtained volume discharge and total N in treated water of each WWTP from Virginia Pollutant Discharge Elimination System and we have included these information in the revised manuscript: Noman Cole WWTP discharges ~140.8 million liters of

water and 370 kg N per day into Pohick Creek. Mooney WWTP discharges ~54.9 million liters of water and 147 kg N per day into the Neabsco Creek. Aquia WWTP discharges much less water and N into the Aquia Creek (~21.2 million liters per day and 35 kg N per day).

We were also able to obtain the river discharges at monitoring stations upstream of the Mooney WWTP (monitor station of Neabsco Creek at Dale City, Virginia) and Aquia WWTP (monitor station of Aquia Creek near Garrisonville, Virginia) from United States Geological Survey (USGS) and compared them to their WWTPs' effluent volumes in order to evaluate the dilution effect on $N_2O$ concentrations and emissions (Figures R1-R3 below). In addition, total nitrogen concentrations were available from the monitor station upstream of Mooney WWTP (Richmond Highway, Virginia). We then compared the total N flow between the Neabsco Creek flow and Mooney WWTP effluent.

[Figure]

Figure R1. Comparison of water flows and total nitrogen inputs from Mooney WWTP effluent and Neabsco Creek. Climatological river flow rates were used for Neabsco Creek because river flow data were not available for years 2022 and 2023.

The volume and nitrogen discharge of Mooney WWTP effluent were always higher than the Neabsco Creek (Figure R1 above). Therefore, the dilution of $N_2O$ in Mooney WWTP effluent by the river flow was small. In contrast, the volume of Aquia WWTP effluent was generally lower than the Aquia Creek flow rate (Figure R2 below). The high dilution by the river flow likely diminished the $N_2O$ signal from Aquia WWTP. In addition, river flow rates were generally lower in summer while WWTPs' effluent volumes were relatively constant throughout the year, leading to a larger ratio of WWTPs' effluent to the river flow (less dilution) in the dry season. That's probably one of reasons why the highest $N_2O$ concentrations were observed downstream Mooney WWTP in August when the river flow was low.

[Figure]

Figure R2. Comparison of water flows from Aquia WWTP effluent and Aquia Creek.

[Figure]

Figure R3. The ratio of WWTP effluent to river flow.

The water column depths of sampling stations have been added to the manuscript: "The embayment stations were 2-3 meters deep while the average depth of central channel stations was around 8 meters". Sedimentary $N_2O$ production may supply $N_2O$ to the water column and further $N_2O$ emissions to the atmosphere. But we don't have direct observations to support that, which deserves further observations.

The bibliographical references cited do not always follow the same criteria (chronological order or alphabetical order).

We have updated the reference order based on the journal's requirement (chronologically in text and alphabetically at the end of the manuscript).

SPECIFIC COMMENTS

Ln 49. In a more recent article than those cited, Rosentreter et al., 2023 there are compiled N2O emissions data from various estuaries, providing a wider range of emissions variation (0.2 – 5.7 Tg N2O yr-1). Specifically, the paper states: "Global estimates of estuarine N2O emissions are highly uncertain, with large discrepancies for both observation-based (220–5,710 GgN2O yr−1) and modelling approaches (94–1,084 GgN2O yr−1).

We have updated the range of estimated estuarine $N_2O$ emissions, citing Rosentreter et al., 2023.

Ln 65-66. References should be listed in ascending chronological order, consistent with the rest of the paper.

References are now cited chronologically in the text.

Ln 87. It should be indicated what type of treatment is given in the WWTPs (primary, secondary, tertiary, etc.) in order to understand if the nitrogen removal capacity of the three wastewater treatment plants is the same. At what distance from the WWTPs were the samples taken? Were the samples taken at approximately the same distance from the discharge point at all three WWTPs? Were the channels where the samples were taken similar? Did they have approximately the same water volume? An important factor when comparing the amount of N2O in the receiving channels is dilution.

All the WWTPs involved in this study implement tertiary treatment. We have listed the distance between the sampling stations and WWTPs: "The distances from the sampling stations to Noman Cole, Mooney, Aquia were approximately 4, 1.8 and 5.8 km, respectively".

See the response to the general comments on the dilution effect.

Ln 99. Were the samples collected from a vessel? Please specify.

Samples were collected on vessel – "Grady White 208", which has been added to the manuscript.

Ln 110-11. How was a 3 mL air headspace created in the 60 mL serum bottles? Did all samples have exactly the same volume of air headspace? Could this 3 mL of air in contact with the sample potentially interfere with the measurement? Was the N2O content in the air also measured? Were the samples taken in duplicate?

Water $N_2O$ concentration samples were collected in triplicate at each sampling sites. 3 mL air headspace was created by removing 3 mL water using a syringe. The monthly atmospheric $N_2O$ concentrations were obtained from the nearby atmospheric station in Brentwood, Maryland (https://gml.noaa.gov/dv/site/?stacode=BWD) (Andrews et al., 2023). The amount of $N_2O$ in 3 mL air headspace was generally less than 4% of the amount of $N_2O$ dissolved in the 57 mL water samples. Thus, the effect of 3 mL air on $N_2O$ measurements was minor and was accounted for the concentration calculations. The similar sampling method has previously been used (e.g., Kelly et al., 2020). We have added this description in the manuscript.

Ln 112. Leave a space between the 10 and the M.

Modified as suggested.

Ln 124-128. Figure 4a, depicting the sampling points of the four streams/rivers (Neabsco Creek (5 stations), Occoquan River (3 stations), Pohick Creek (4 stations), and Accotink Creek), should be included in the Materials and Methods section.

Rather than cite Figure 4 out of order in the text, or clutter up Figure 1 (we tried that, it makes the figure unreadable at the necessary scale), we now cite Figure 4 in the caption of Figure 1 for the locations of the additional creek sampling stations.

Ln 128. Where have the data on water discharge and nitrogen (kg) per day from the wastewater treatment plants been obtained? It would be interesting to include this information in the manuscript.

Data source has been added: "We obtained volume discharge and total N in treated water of each WWTP from Discharge Monitoring Reporting required by Virginia Pollutant Discharge Elimination System permit".

Ln 149. It is not reflected in the text how the 3 mL air headspace is taken from the serum bottles to estimate the amount of N2O in the sample.

See the response to the related comment above.

Ln 167. "The equilibrium N2O concentration was calculated based on the solubility of N2O (Weiss and Price, 1980)…" Where did you obtain the value of N2O in the atmosphere for the calculations? Which value did you consider, the daily, monthly...?

The monthly atmospheric $N_2O$ concentrations were obtained from the nearby atmospheric station in Brentwood, Maryland (https://gml.noaa.gov/dv/site/?stacode=BWD) (Andrews et al., 2023).

Ln 170: What do the initials NCEP stand for? It would be more comprehensive to include the website from which the value of U was taken.

NCEP stands for National Centers for Environmental Prediction and the website of the data source has been added to the manuscript (https://psl.noaa.gov/data/gridded/data.ncep.reanalysis.html).

Ln 171. You should cite in the paper the expression from which Sc has been estimated, possibly from the proposed expression by Wanninkhof (2014). You should indicate whether for the calculation of Sc, you have considered the expression for salinity equal to zero, or if, on the contrary, the N2O Schmidt number for each point has been scaled to the values proposed by Wanninkhof (2014) for salinities between 0 and 35, assuming that Sc varies linearly with salinity.

We have added in the text: "Schmidt number was estimated as a function of temperature based on the equation from Wanninkhof (2014). Since our samples have salinity close to 0, we used the parameterization for freshwater".

Ln 173. References should be listed in ascending chronological order, consistent with the rest of the paper.

References order has been updated.

Ln 170-176. I don't understand why they are using a gas transfer velocity parameterization (k) proposed for the ocean, such as the expression by Wanninkhof (2014), rather than a k for a coastal system. If they didn't have data on current velocity and depth of the system necessary to use the k by Borges et al. (2004) and Rosentreter et al. (2021), they could have used the expression proposed by Raymond and Cole (2001), which is based on a compilation of k proposed for different coastal systems, or that by Jiang et al. (2008), based on the compilation of Raymond and Cole (2001) as well as other studies conducted in estuaries. Furthermore, given the uncertainty associated with k, to minimize this, they could have estimated water-atmosphere fluxes considering two expressions of k (Raymond and Cole, 2001; Jiang et al., 2008; Wanninkhof, 2014), and taken the average value of the three fluxes obtained, as many other authors do in coastal systems (e.g., Call et al., 2015; Sánchez-Rodriguez et al., 2024).

This is a great suggestion. Following the reviewer's comment, we have now estimated N$_2$O fluxes based on three different parameterizations of k values (Raymond and Cole, 2001; Jiang et al., 2008; Wanninkhof, 2014).

$k_{600} = 1.91 \times e^{0.35 \times U}$ (Raymond and Cole, 2001)
$k_{600} = 0.314 \times U^2 - 0.436 \times U + 3.99$ (Jiang et al., 2008)
$k = k_{600} \times \left(\frac{Sc}{600}\right)^{-0.5}$
$k = 0.251 \times U^2 \times (\frac{Sc}{660})^{-0.5}$ (Wanninkhof, 2014)

Average values of the three estimates are presented in the manuscript and estimates of each parameterization are provided in the associated dataset.

Ln 233. I suggest wording it like this: ....vs 6‰ for stations of the central channel and without the influence by WWTPs

Modified the text as suggested.

Ln 242. In general, denitrification typically occurs in environments with low oxygen concentrations (DO ≤ 5 μmol L−1, Codispoti et al., 2001). As illustrated in Supplementary Figure 2, oxygen concentrations at the stations never reached low values. In fact, downstream stations of wastewater treatment plants exhibited dissolved oxygen levels ranging between 139.38 (25/07/2022) – 430.94 μM (7/02/2023). It is recognized that denitrification can also take place within oxygenated water columns containing suspended organic matter particles (Bange, 2008). Is there a substantial amount of suspended material in the studied system that could

induce denitrification in oxygenated water? On the other hand, it is well-established that coastal sediments provide optimal environments for denitrification due to continuous inputs of nutrients and organic matter from land. Could it be that some of the measured N2O in the water originates from the sediment?

There was a substantial amount of suspended material in the study region: the total suspended particle concentration was 14.8±10 mg/L and the Secchi depth was generally below 1 m. We don't have direct evidence to show but acknowledge the possibility that denitrification could occur in the anoxic particles or in the sediments, supplying $N_2O$ to the water column.

Ln 252. Correlations of 0.62 (r2=0.38) and 0.51 (r2=0.26), I do not consider them strong correlations, remove the word strong.

"Strong" was removed from the text.

Ln 252-256 and 263-264. In stations unaffected by WWTPs, there appears to be a good positive correlation between N2O and DO and NOx, which could indicate that nitrification is an important process in these N2O production stations.

We have now added in the text: "Although previous studies have showed dissolved oxygen to be an important driver of $N_2O$ concentrations or fluxes in rivers and estuaries (Rosamond et al., 2012; Wang et al., 2015; Zheng et al., 2022), we did not find a strong dependence of $N_2O$ on oxygen concentrations in the Potomac River Estuary (Figure 3a). This lack of strong dependence is probably because of the overall oxygenated conditions (Supplementary Figure 1c) and opposite correlations found in stations without WWTPs (positive) or with WWTPs (negative) (Supplementary Figure 3), which could lead to different $N_2O$ production pathways".

Ln 262. References should be listed in ascending chronological order, consistent with the rest of the paper.

References order has been updated.

Ln 278-279. References should be listed in ascending chronological order, consistent with the rest of the paper.

References order has been updated.

Ln 292-294. Why does it not also present the predictive model of N2O concentration based on total nitrogen and temperature for stations in the central channel of the Potomac Estuary? It could be interesting to have it to extrapolate to other areas of the estuary located in the channel. Perhaps you have included the data measured in the channel in the samples without wastewater treatment plants (WWTPs). If so, please indicate it. I believe you should have stated the number of stations/data considered in each prediction.

We have clarified in the text: "Predictions were performed separately for stations with WWTPs ($N_2O\ concentration = 0.115 \times total\ N - 0.241 \times temperature + 17.185$, n=18, r=0.78;

p<0.01) and without WWTPs including central channel stations ($N_2O$ $concentration$ = $0.049 \times total\ N - 0.298 \times temperature + 18.888$, n=23, r=0.81, p<0.01)".

Ln 298-300. Did you use the prediction model for stations without WWTPs? Please indicate it in the paper.

The embayment station in the Occoquan River was not in the downstream of WWTPs. $N_2O$ concentrations were estimated using the predictive model built upon stations without WWTPs. For comparison, we have now made predictions for another station in the Pohick Creek that is downstream of Noman Cole WWTP, using the predictive model built upon stations with WWTPs. (see Figures R4-R5 below).

[Figure]

Figure R4. Historical measurements of temperature (a) and N concentration (b) at the Occoquan Bay sampling station without the influence of WWTPs. $N_2O$ concentration (c) is predicted based on a multiple linear regression model developed for stations without the influence from WWTPs. The red points are the observed $N_2O$ concentration.

[Figure]

Figure R5. Historical measurements of temperature (a) and N concentration (b) at the Pohick Bay sampling station with the influence of Noman Cole WWTP. $N_2O$ concentration (c) is predicted based on a multiple linear regression model developed for stations with the influence from WWTPs. The red points are the observed $N_2O$ concentrations.

FIGURES

Figure 2 and Supplementary Figure 1. What does the "01" after the slash indicate on the x-axis of the central graphs? Wouldn't it be more intuitive for the reader to use "22" or "23" instead of "01," depending on the year the sampling was conducted?

We have now changed the axis tick labels to the format "year/month" as suggested.

Supplementary Figure 2. In the figure caption and in Figure a, a negative sign as a subscript is missing on NOx- on the x-axis. In Figure b, on the x-axis, remove the space between N2 and O.

Figure text and captions have been modified as suggested.

REFERENCES:

Bange, H. W., 2008. Gaseous Nitrogen Compounds (NO, N2O, N2, NH3) in the Ocean. In Nitrogen in the Marine Environment (pp. 51–94). Elsevier Inc. https://doi.org/10.1016/B978-0-12-372522-6.00002-5

Call, M., Maher, D. T., Santos, I. R., Ruiz-Halpern, S., Mangion, P., Sanders, C. J., ... & Eyre, B. D., 2015. Spatial and temporal variability of carbon dioxide and methane fluxes over semidiurnal and spring–neap–spring timescales in a mangrove creek. Geochimica et Cosmochimica Acta, 150, 211-225.

Codispoti, L.A., Brandes, J.A., Christensen, J.P., Devol, A.H., Naqvi, S.W.A., Paerl, H.W., Yoshinari, T., 2001. The oceanic fixed nitrogen and nitrous oxide budgets: moving targets as we enter the anthropocene? Sci. Mar. 65 (S2), 85–105.

Jiang, L.Q., Cai, W.J., & Wang, Y. (2008). A comparative study of carbon dioxide degassing in river- and marine-dominated estuaries. Limnology and Oceanography, 53(6), 2603–2615.

Rosentreter, J. A., Laruelle, G. G., Bange, H. W., Bianchi, T. S., Busecke, J. J. M., Cai, W. J., Eyre, B. D., Forbrich, I., Kwon, E. Y., Maavara, T., Moosdorf, N., Najjar, R. G., Sarma, V. V. S. S., Van Dam, B., & Regnier, P., 2023. Coastal vegetation and estuaries are collectively a greenhouse gas sink. Nature Climate Change, 13(6), 579–587. https://doi.org/10.1038/s41558-023-01682-9

Sánchez-Rodríguez, J., Ortega, T., Sierra, A., Mestre, M., Ponce, R., Fernández-Puga, M. D. C., & Forja, J., 2024. Distribution, reactivity and vertical fluxes of methane in the Guadalquivir Estuary (SW Spain). Science of The Total Environment, 907, 167758.

References:

Andrews, A., Crotwell, A., Crotwell, M., Handley, P., Higgs, J., Kofler, J., Lan, X., Legard, T., Madronich, M., McKain, K., Miller, J., Moglia, E., Mund, J., Neff, D., Newberger, T., Petron, G., Turnbull, J., Vimont, I., Wolter, S., & NOAA Global Monitoring Laboratory. (2023). NOAA Global Greenhouse Gas Reference Network Flask-Air PFP Sample Measurements of N2O at Tall Tower and other Continental Sites, 2005-Present [Data set]. NOAA GML. https://doi.org/10.15138/C11N-KD82 Version: 2023-08-23.

de Haas, D., & Andrews, J. (2022). Nitrous oxide emissions from wastewater treatment-Revisiting the IPCC 2019 refinement guidelines. *Environmental Challenges*, *8*, 100557.

Jiang, L. Q., Cai, W. J., & Wang, Y. (2008). A comparative study of carbon dioxide degassing in river-and marine-dominated estuaries. *Limnology and Oceanography*, *53*(6), 2603-2615.

Raymond, P. A., & Cole, J. J. (2001). Gas exchange in rivers and estuaries: Choosing a gas transfer velocity. *Estuaries*, *24*(2), 312-317.

Rosentreter, J. A., Laruelle, G. G., Bange, H. W., Bianchi, T. S., Busecke, J. J., Cai, W. J., ... & Regnier, P. (2023). Coastal vegetation and estuaries are collectively a greenhouse gas sink. *Nature Climate Change*, *13*(6), 579-587.

Wanninkhof, R.: Relationship between wind speed and gas exchange over the ocean revisited, Limnol. Oceanogr. Methods, 12, 351-362, 2014.

Zhao, Y. W., Du, L. L., Hu, B., Lin, H. Y., Liang, B., Song, Y. P., ... & Wang, H. C. (2024). Impact of influent characteristics and operational parameters on nitrous oxide emissions in wastewater treatment: Strategies for mitigation and microbial insights. *Current Research in Biotechnology*, 100207.

---

## Author Comment (AC2)

Reviewer 2:
The manuscript "Variable contribution of wastewater treatment plan effluents to nitrous oxide emission" by Tang et al. studies the effects of wastewater treatment plants on the Potomac River estuary in the United States. For over one year, they took monthly samples for nitrous oxide, total nitrogen and dissolved inorganic nitrogen concentrations. Generally, the results showed spatial and seasonal variability in nitrous oxide concentrations with higher concentrations downstream of the WWTPs, highlighting the importance of WWTPs regarding estuarine N2O emissions. Therefore, this manuscript will be of interest in the context of global N2O emissions from estuaries and WWTPs. The data set is well presented and interpreted and the text well written and organized. However, major revisions are necessary to discuss effects of wastewater treatment processes and dilution effects.

General remarks:
The paper misses to discuss differences in wastewater treatments and dilution effects, which leads to some important unanswered questions:

- Do the WTTPs differ in type, removal strategy and treated water volume? Are differences visible in TN, DIN and N2O effluents?

- How big are the water volumes of the WTTP effluents compared to the water volume in the estuary (especially in the tributaries)? I would recommend calculating a wastewater discharge fraction of stream flow.

- How big is the N load in the WTTP effluents compared to the N loads in the upstream river? How are the effluents diluted and are concentration increases expected/seen?

- Are there seasonal effects on the impact of wastewater effluents? For example, Murray et al. (2020) measured differences in N2O concentrations affected by WWTPS between dry and wet season in an Australian estuary.

Thank the reviewer for their great comments on differences in the wastewater treatment processes among WWTPs and the dilution effect on $N_2O$ concentrations by riverine discharge. The first reviewer also had the similar comments. We have responded to the reviewer's comments below in blue font and made changes accordingly in the manuscript.

Although we contacted the WWTPs directly, we were not able to obtain detailed information about the treatment processes of the three treatment plants except they all implement tertiary treatment. We acknowledged that the different types of treatment affect the $N_2O$ production yield in the WWTPs in the text (de Haas and Andrews. 2022; Zhao et al., 2024).

For evaluating the dilution effect, we were able to obtain the volume discharge and total N in treated water of each WWTP from Virginia Pollutant Discharge Elimination System and have included that information in the revised manuscript: Noman Cole WWTP discharges ~140.8 million liters of water and 370 kg N per day into Pohick Creek. Mooney WWTP discharges ~54.9 million liters of water and 147 kg N per day into the Neabsco Creek. Aquia WWTP

discharges much less water and N into the Aquia Creek (~21.2 million liters per day and 35 kg N per day).

We were also able to obtain the river discharges at monitoring stations upstream of the Mooney WWTP (monitor station of Neabsco Creek at Dale City, Virginia) and Aquia WWTP (monitor station of Aquia Creek near Garrisonville, Virginia) from United States Geological Survey (USGS) and compared them to their WWTPs' effluent volumes in order to evaluate the dilution effect on $N_2O$ concentrations and emissions (Figures R1-R3). In addition, total nitrogen concentrations were available from the monitor station upstream of Mooney WWTP (Richmond Highway, Virginia). We then compared the total N flow between the Neabsco Creek flow and Mooney WWTP effluent.

[Figure]

Figure R1. Comparison of water flows and total nitrogen inputs from Mooney WWTP effluent and Neabsco Creek.

The volume and nitrogen discharge of Mooney WWTP effluent were always higher than the Neabsco Creek (Figure R1 above). Therefore, the dilution of $N_2O$ in Mooney WWTP effluent by the river flow was small. In contrast, the volume of Aquia WWTP effluent was generally lower than the Aquia Creek flow rate (Figure R2 below). The high dilution by the river flow likely diminished the $N_2O$ signal from Aquia WWTP. In addition, river flow rates were generally lower in summer while WWTPs' effluent volumes were relative constant throughout the year, leading to a larger contribution of WWTPs' effluents to total river flow (less dilution) in the dry season. That's probably one of reasons why the highest $N_2O$ concentrations were observed downstream Mooney WWTP in August. This is similar to what Murry et al. (2020) found in an Australian estuary as the reviewer pointed out.

[Figure]

Figure R2. Comparison of water flows from Aquia WWTP effluent and Aquia Creek.

[Figure]

Figure R3. The ratio of WWTP effluent to river flow. The horizontal dashed line denotes a ratio of 1.

Specific remarks:
L63: "[…] are highly variable, and are normally […]"

Text has been modified as suggested.

L75: What is the mean annual discharge entering the estuary from the upstream river? What are mean N loads?

We have added in the text: "Potomac River discharge has been measured by the USGS at Chain Bridge near Washington, DC. The annual mean discharge from 1895 to 2002 at Chain Bridge was 321 $m^3$ $s^{-1}$ with a large interannual variability (Jaworski et al., 2007). The annual total nitrogen loading is estimated to be around 27.7 $\times 10^6$ kg N $year^{-1}$ in 2008-2009 (Bricker et al., 2014)".

L84: "[…] nitrogen effluent concentration below 7.5 mg L-1 […]"

Text has been clarified.

L108: At what tidal state was the sampling carried out? How does the tidal state affect the results? Did you always sampled at the same tidal state to minimize effects?

Since the routine water quality sampling by the Department of Environmental Quality generally occurs in the morning, we were not able to collect samples at the same tidal state. We acknowledge this caveat in the text: "While estuarine $N_2O$ concentrations could be affected by tides (Gonçalves et la., 2015), sampling was not always conducted at the same tidal state due to logistic difficulties".

L110: Did you take replicates?

Yes, triplicate DIN concentration and $N_2O$ samples were collected.

L110-111: Did you measure N2O concentrations in air headspace for correction? How did you estimate/measure atmospheric N2O concentrations?

The monthly atmospheric $N_2O$ concentrations were obtained from the nearby atmospheric station in Brentwood, Maryland (https://gml.noaa.gov/dv/site/?stacode=BWD) (Andrews et al., 2023).

L151: Did you measure replicates for N2O isotopes?

Yes, this is now clarified in the text.

L169-170: Why did you decide to use Wannikhof's formula, which applies better to open oceans? There are formulas specifically designed for estuarine environments, e.g. Clark et al. (1995) and Raymond and Cole (2001).

Following the comments from both reviewers, we have used three different parametrizations (Raymond and Cole, 2001; Jiang et al., 2008; Wanninkhof, 2014) to calculate gas transfer coefficient to estimate $N_2O$ fluxes. Average values of thee three estimates are presented in the manuscript and estimate of each parameterization is provided in the associated dataset.

$k_{600} = 1.91 \times e^{0.35 \times U}$ (Raymond and Cole, 2001)
$k_{600} = 0.314 \times U^2 - 0.436 \times U + 3.99$ (Jiang et al., 2008)
$k = k_{600} \times \left(\frac{Sc}{600}\right)^{-0.5}$
$k = 0.251 \times U^2 \times (\frac{Sc}{660})^{-0.5}$ (Wanninkhof, 2014)

L128-131: How do these values (treated water volumes and N loads) compared to the riverine volume and N loads? See general comments above. Did you see changing impacts depending on the size of the WWTPs?

See the reply to the general comments above.

L149: How did you take the amount of N2O in the 3 mL headspace into account?

We have clarified in the text: 3 mL air headspace was created by removing 3 mL water using a syringe.

The monthly atmospheric $N_2O$ concentrations were obtained from the nearby atmospheric station in Brentwood, Maryland (https://gml.noaa.gov/dv/site/?stacode=BWD) (Andrews et al., 2023). The amount of $N_2O$ in 3 mL air headspace was generally less than 4% of the amount of N2O dissolved in the 57 mL water samples. Thus, the effect of 3 mL air on $N_2O$ measurements was minor and was accounted for the concentration calculations.

L171: How did you calculate the Schmidt number?

We have added in the text: "Schmidt number was estimated as a function of temperature based on the equation from Wanninkhof (2014). Since our samples have salinity close to 0, we used the parameterizations for freshwater."

L185: Do you also see these seasonal differences in the effect of the WWTPs? The effluent of WWTPs usually have a relatively constant N load throughout the entire year. Therefore, I could imagine that it makes a big difference whether the WWTPs discharge into an estuary with a high N concentration in winter or a low N concentration in summer. Further, riverine discharge is usually higher in winter, which leads to greater dilution and reduces the impact of WWTP effluents.

The reviewer is correct about the seasonal changes in the volume of WWTPs' effluent vs the riverine discharge. See response to the general comments above.

L190-191: Does this also reflects in seasonal changing δ15N-N2O values?

Yes, we saw a seasonal change in δ15N-N2O at stations downstream of WWTPs. We have added in the text: "$\delta^{15}N$ of $N_2O$ for stations with the influence of WWTPs showed a clear seasonal variation: higher values in summer than winter (Figure 2e). This seasonal difference may be related to the seasonal change in the relative importance of WWTPs' effluents versus riverine discharge (Supplementary Figure 2c). For example, relatively larger WWTPs' effluents led to larger $\delta^{15}N$ of $N_2O$ in summer when riverine flows were smaller. However, no clear seasonal pattern of $\delta^{15}N$ of $N_2O$ was seen for stations without the influence of WWTPs".

L218: Calculating a wastewater discharge fraction of stream flow would help to estimate the different dilution effects for each WWTP.

See response to the general comments above.

L220: Can you estimate the wastewater discharge fraction of stream flow considering the water volume of the estuary and water volume and N load from the WWTP?

See response to the general comments above.

L224: "High-resolution spatial and temporal sampling" – I don't agree that the conducted sampling campaign has a high spatial and temporal resolution considering the existence of laser-based measurements that allow resolution by the second. Sampling was conducted once or twice a month at eleven stations or once at 14 stations. I would suggest rephrasing this statement.

We have modified the text to: "Repeated spatial and temporal sampling allowed us to capture these $N_2O$ hotspots".

L233: Do you observe seasonal changes?

Yes, see the response to the related comments above.

L238: What kind of treatments are performed at the WWTPs discharging into the Potomac River estuary? There are different ways of operating N removal within WWTP (biological, chemical, and physical methods) (e.g. Winkler and Straka, 2019; Zhou et al., 2023). Further, biological removal strategies, for example, can also differ significantly: (1) denitrification followed by nitrification, where a part of the treated water is fed back into the denitrification after nitrification, (2) nitrification is followed by denitrification with organic carbon being added to the denitrification chamber (e.g. part of the untreated water before nitrification), (3) intermittent denitrification, in which longer phases with aerobic nitrification and anoxic denitrification alternate in the same tank, (4) simultaneous denitrification due to the discontinuous or punctual supply of oxygen, (5) cascade denitrification, in which the wastewater passes through several tanks with alternating denitrification and nitrification, or (6) alternating denitrification, consisting of two aeration tanks that are alternately fed with wastewater and aerated. N2O production and N2O production pathway may differ significantly depending on the treatment strategy. Therefore, it would be very valuable to discuss treatment strategies considering possible isotope changes. Do the WWTP even use biological treatments or other physical/chemical ones?

We agree with the reviewer that the type of treatment affects the nitrogen removal efficiency and $N_2O$ production yield in the WWTPs (de Haas and Andrews. 2022; Zhao et al., 2024). However, the lack of information about the types of treatment process of WWTPs in this study prevented us from comparing their $\delta^{15}N$ of $N_2O$ values. Thus, we focused on the spatiotemporal variation in $\delta^{15}N$ of $N_2O$.

L242: Oxygen concentration during your measurements (supplementary material Fig. 1, L264) were always above the threshold for denitrification ($< 6.25$ μM; Seitzinger, 1988). Denitrification can occur in anoxic microsites close to particles (Liu et al., 2013; Zhu et al., 2018; Schulz et al., 2022) or in anoxic sediments. Where do you suggest denitrification occurs? Is it an artefact of denitrification in the WWTP?

We are not certain about the locations of the denitrification. As the reviewer pointed out, denitrification could occur in anoxic zones in particles or sediments. $N_2O$ close to WWTPs' effluents had elevated $\delta^{15}N$ values compared to upstream stations (Figure 4b) suggested that at

least part of $N_2O$ consumption occurred in the WWTPs. The $\delta^{15}N$ values of $N_2O$ could be modified by $N_2O$ cycling processes downstream WWTPs including denitrification in anoxic particles and sediments.

L250: Not a strong (r = 0.51), but a significant correlation (p<0.01) – Thus, I would rephrase "N2O concentrations showed a significant positive correlation […]"

Text has been modified.

L254: Did you observe correlations between NH4+ and/or NO2- concentrations with N2O?

$NH_4^+$ and $NO_2^-$ concentrations were measured at a few selected stations. Their concentrations were much smaller than $NO_3^-$ alone, mostly accounting for less than 10% of the DIN concentration. In addition, there was no clear correlations between $NH_4^+$ and $N_2O$ or $NO_2^-$ and $N_2O$ (see Figure R4 below).

[Figure]

Figure R4. Relationship between $N_2O$ and $NH_4^+$ concentrations, and between $N_2O$ and $NO_2^-$ concentrations.

Figure 3: Why is Chlorophyll a in brackets?

We have changed to use the full name of Chlorophyll a in the figure.

L292: "WWTPs"

Modified.

L299: Did you use the prediction with or without WWTPs?

The embayment station in the Occoquan River was not in the downstream of WWTPs. Thus, we used the predictive model built upon stations without WWTPs. For comparison, we have now made predictions for another station in the Pohick Creek that is downstream of Noman Cole WWTP, using the predictive model built upon stations with WWTPs. (see Figures R5-R6 below).

[Figure]

Figure R5. Historical measurements of temperature (a) and N concentration (b) at the Occoquan Bay sampling station without the influence of WWTPs. $N_2O$ concentration (c) is predicted based on a multiple linear regression model developed for stations without the influence from WWTPs. The red points are the observed $N_2O$ concentration.

[Figure]

Figure R6. Historical measurements of temperature (a) and N concentration (b) at the Pohick Bay sampling station with the influence of Noman Cole WWTP. $N_2O$ concentration (c) is predicted based on a multiple linear regression model developed for stations with the influence from WWTPs. The red points are the observed $N_2O$ concentrations.

L317: Did you consider tidal state during your sampling (e.g. always sampled at same tidal state)?

We clarified in the text: "While estuarine $N_2O$ concentrations could be affected by tides (Gonçalves et la., 2015), sampling was not always conducted at the same tidal state due to logistic difficulties".

L334-335: Remove space between "NOx-" and ","

There was no space between NOx- and ",". It was because of the different lines.

L357-359: Brown et al. (2022) also found estuarine type, mixing regime and stratification important factors controlling N2O emissions.

We have added these factors and cited Brown et al. (2022) in the text.

Supplementary Material S24: "δ15N of NOx concentration (a) and N2O concentration (b)"

Caption of this Supplementary Figure has been clarified: "The change in $\delta^{15}N$ of $N_2O$ in relation to the changes in $NO_x^-$ concentrations (a) and $N_2O$ concentrations (b)."

Supplementary Material Fig. 3: Why is Chlorophyll a in brackets?

Full name of Chlorophyll a is now shown in the figure.

Supplementary Material L33: "[…] the influence of WWTPs […]"

Text was modified as suggested.

References

Brown, A. M., Bass, A. M., and Pickard, A. E.: Anthropogenic-estuarine interactions cause disproportionate greenhouse gas production: A review of the evidence base, Mar. Pollut. Bull., 174, 113240, https://doi.org/10.1016/j.marpolbul.2021.113240, 2022.

Clark, J. F., Schlosser, P., Simpson, H. J., Stute, M., Wanninkhof, R., and Ho, D. T.: Relationship between gas transfer velocities and wind speeds in the tidal Hudson River determined by the dual tracer technique, in: Air-Water Gas Transfer, edited by: Jähne, B. and Monahan, E. C., AEON Verlag, Hanau, 785–800, 1995.

Liu, T., Xia, X., Liu, S., Mou, X., and Qiu, Y.: Acceleration of denitrification in turbid rivers due to denitrification occurring on suspended sediment in oxic waters, Environ. Sci. Technol., 47, 4053–4061, https://doi.org/10.1021/es304504m, 2013.

Murray, R. H., Erler, D. V., Rosentreter, J., Wells, N. S., and Eyre, B. D.: Seasonal and spatial controls on N2O concentrations and emissions in low-nitrogen estuaries: Evidence from three tropical systems, Mar. Chem., 221, 103779, https://doi.org/10.1016/j.marchem.2020.103779, 2020.

Raymond, P. A. and Cole, J. J.: Gas exchange in rivers and estuaries: Choosing a gas transfer velocity, Estuaries, 24, 312–317, https://doi.org/10.2307/1352954, 2001.

Schulz, G., Sanders, T., van Beusekom, J. E. E., Voynova, Y. G., Schöl, A., and Dähnke, K.: Suspended particulate matter drives the spatial segregation of nitrogen turnover along the hyper-turbid Eestuary, Biogeosciences, 19, 2007–2024, https://doi.org/10.5194/bg-19-2007-2022, 2022.

Seitzinger, S. P.: Denitrification in freshwater and coastal marine ecosystems: Ecological and geochemical significance, Limnol. Oceanogr., 33, 702–724, https://doi.org/10.4319/lo.1988.33.4part2.0702, 1988.

Winkler, M. K. H. and Straka, L.: New directions in biological nitrogen removal and recovery from wastewater, Curr. Opin. Biotechnol., 57, 50–55, https://doi.org/10.1016/j.copbio.2018.12.007, 2019.

Zhou, Y., Zhu, Y., Aber, J., Li, C., and Chen, G.: A Comprehensive Review on Wastewater Nitrogen Removal and Its Recovery Processes, Int. J. Environ. Res. Public. Health, 20, 3429, https://doi.org/10.3390/ijerph20043429, 2023.

Zhu, W., Wang, C., Hill, J., He, Y., Tao, B., Mao, Z., and Wu, W.: A missing link in the estuarine nitrogen cycle?: Coupled nitrification-denitrification mediated by suspended particulate matter, Sci. Rep., 8, 2282, https://doi.org/10.1038/s41598-018-20688-4, 2018.

References:

Andrews, A., Crotwell, A., Crotwell, M., Handley, P., Higgs, J., Kofler, J., Lan, X., Legard, T., Madronich, M., McKain, K., Miller, J., Moglia, E., Mund, J., Neff, D., Newberger, T., Petron, G., Turnbull, J., Vimont, I., Wolter, S., & NOAA Global Monitoring Laboratory. (2023). NOAA Global Greenhouse Gas Reference Network Flask-Air PFP Sample Measurements of N2O at Tall Tower and other Continental Sites, 2005-Present [Data set]. NOAA GML. https://doi.org/10.15138/C11N-KD82 Version: 2023-08-23.

Bricker, S. B., Rice, K. C., and Bricker, O. P.: From Headwaters to Coast: Influence of Human Activities on Water Quality of the Potomac River Estuary, Aquatic Geochemistry, 20, 291-323, 10.1007/s10498-014-9226-y, 2014.

Brown, A. M., Bass, A. M., & Pickard, A. E. (2022). Anthropogenic-estuarine interactions cause disproportionate greenhouse gas production: A review of the evidence base. *Marine Pollution Bulletin*, *174*, 113240.

de Haas, D., & Andrews, J. (2022). Nitrous oxide emissions from wastewater treatment-Revisiting the IPCC 2019 refinement guidelines. *Environmental Challenges*, *8*, 100557.

Gonçalves, C., Brogueira, M. J., & Nogueira, M. (2015). Tidal and spatial variability of nitrous oxide (N2O) in Sado estuary (Portugal). *Estuarine, Coastal and Shelf Science*, *167*, 466-474.

Jaworski, N. A., Romano, B., Buchanan, C., and Jaworski, C.: The Potomac River Basin and its Estuary: landscape loadings and water quality trends, 1895–2005, Report, Interstate Commission on the Potomac River Basin, Rockville, Maryland, USA, 2007.

Jiang, L. Q., Cai, W. J., & Wang, Y. (2008). A comparative study of carbon dioxide degassing in river-and marine-dominated estuaries. *Limnology and Oceanography*, *53*(6), 2603-2615.

Kelly, C. L., Travis, N. M., Baya, P. A., and Casciotti, K. L.: Quantifying Nitrous Oxide Cycling Regimes in the Eastern Tropical North Pacific Ocean With Isotopomer Analysis, Global Biogeochemical Cycles, 35, 10.1029/2020gb006637, 2021.

Murray, R., Erler, D. V., Rosentreter, J., Wells, N. S., & Eyre, B. D. (2020). Seasonal and spatial controls on N2O concentrations and emissions in low-nitrogen estuaries: Evidence from three tropical systems. *Marine Chemistry*, *221*, 103779.

Raymond, P. A., & Cole, J. J. (2001). Gas exchange in rivers and estuaries: Choosing a gas transfer velocity. *Estuaries*, *24*(2), 312-317.

Rosamond, M. S., Thuss, S. J., and Schiff, S. L.: Dependence of riverine nitrous oxide emissions on dissolved oxygen levels, Nature Geoscience, 5, 715-718, 10.1038/ngeo1556, 2012.

Wang, J., Chen, N., Yan, W., Wang, B., and Yang, L.: Effect of dissolved oxygen and nitrogen on emission of $N_2O$ from rivers in China, Atmospheric Environment, 103, 347-356, 10.1016/j.atmosenv.2014.12.054, 2015.

Wanninkhof, R.: Relationship between wind speed and gas exchange over the ocean revisited, Limnol. Oceanogr. Methods, 12, 351-362, 2014.

Zhao, Y. W., Du, L. L., Hu, B., Lin, H. Y., Liang, B., Song, Y. P., ... & Wang, H. C. (2024). Impact of influent characteristics and operational parameters on nitrous oxide emissions in wastewater treatment: Strategies for mitigation and microbial insights. *Current Research in Biotechnology*, 100207.

Zheng, Y., Wu, S., Xiao, S., Yu, K., Fang, X., Xia, L., Wang, J., Liu, S., Freeman, C., and Zou, J.: Global methane and nitrous oxide emissions from inland waters and estuaries, Glob Chang Biol, 28, 4713-4725, 10.1111/gcb.16233, 2022.

---

## Referee Report (RR1)

The paper by Tang et al., with the title: "Variable contribution of wastewater treatment plant effluents to $N_2O$ emission " has greatly improved in clarity and quality with the new changes. Given that there are very few studies worldwide on the impact of WWTPs on $N_2O$ emissions in aquatic systems, as shown in the Supplementary figure 8 and in the Table 1 of the paper, I consider the article to be of great interest and that it will be widely disseminated. However, I have some suggestions:

**General comments:**

The title of the paper refers to the contribution of WWTPs to $N_2O$ emissions. However, the manuscript hardly discusses water-atmosphere fluxes of $N_2O$, nor the contribution of WWTPs to water-atmosphere fluxes of $N_2O$ in the Potomac River estuary in detail. So my suggestion is that just as the contribution of WWTPs to $N_2O$ concentration is discussed, there should be more discussion of the effect of WWTPs on $N_2O$ fluxes to the atmosphere in the system. Another option would be to change the title of the paper.

**Material and methods:**

Ln 158: There is no mention of phosphate in the manuscript, so you should delete it. Were the samples taken in triplicate like the $N_2O$ samples? Please indicate.

Ln 167-168: Were obtained $N_2O$ concentrations in the water from the measurements made in the headspace using the solubility proposed by Weiss and Price (1980)?

Ln 168-170: The text: "Specifically, the monthly atmospheric $N_2O$ concentrations were obtained from the nearby atmospheric station in Brentwood, Maryland (https://gml.noaa.gov/) (Andrews et al., 2023)." should be included in the $N_2O$ flux calculation section, it could go on line 191 after (Weiss and Price, 1980).

Ln 173: It should be included how you have calculated the saturation percentage. This parameter is discussed in the text and presented in figure 2a.

Ln 186: It should be indicated how the total N cited in lines 158-160 has been measured.

Ln 191-194. The three gas transfer velocity (k) equations should be written in the same format:

- - Or write the k proposed by Wanninkhof (2014) as $k_{660}$ as has been done for the other two parameterisations.:

$$k_{660} = 0.251 \ x \ U^2$$

- - Or write:

Raymond and Cole (2001): $k = 1.91 \ x \ e^{0.35xU} \ x \ \left(\frac{Sc}{600}\right)^{-0.5}$

Jiang et al. (2008): $k = 0.314 \ x \ U^2 - 0.436 \ x \ U + 3.99 \ x \ \left(\frac{Sc}{600}\right)^{-0.5}$

Wanninkhof (2014): $k = 0.251 \ x \ U^2 \ x \ \left(\frac{Sc}{660}\right)^{-0.5}$

**Results and discussion**

Ln 219-228. More should be commented on the water-atmosphere $N_2O$ fluxes, practically only their range of variation in the whole system is presented. As with the $N_2O$ concentrations, the water-atmosphere fluxes present seasonal variations (this if is commented in the abstract) and surely present spatial variations (you should comment on this). However, it is mentioned in the paper that the saturation percentage of $N_2O$ is always higher than 100%, so the system behaves as a source of this gas, and that there is seasonal variation, but little is said about the fluxes to the atmosphere (Ln 218-220).

Ln 226-228: I do not believe that a maximum flux of $N_2O$ to the atmosphere of $31.7 \mu mol$ $m^{-2} d^{-1}$ in the Potomac River Estuary can be considered as an intense source of $N_2O$ to the atmosphere, as there are other estuaries with much more intense emissions. Perhaps it would be more accurate to put: Therefore, tributaries to the Chesapeake Bay (i.e., the Potomac River) are more intense sources of $N_2O$ to the atmosphere than the Bay.

Ln 285-286: For better clarity and interpretation of the text, the values of your observed $\delta^{15}N$ of $N_2O$ downstream of WWTPs and in the urban WWTPs should be included.

Ln 288: Do you know of any work where there is evidence of denitrification in WWTPs, in downstream creeks, or in sediments? If so, could you please cite it.

Ln 343-345: Text in brackets is not in Times New Roman 12. Why is the number of data considered for the predictions so small? Especially for the stations without WWTPs, in the complete study there are 8 sampling x 8 stations (4 stations without WWTPs + 4 stations central channel) = 64 data compared to the 23 considered in the prediction.

Ln 365. In the section: "Impact of wastewater treatment plants on $N_2O$ concentrations and emissions" very little is mentioned about how $N_2O$ fluxes to the atmosphere vary in the stations upstream and downstream of the WWTPs. However, there is much discussion of the effect of the WWTPs on $N_2O$ concentrations. More should be said about these emissions, as the title of the paper says "Variable contribution of wastewater treatment plant effluents to $N_2O$ emission". Furthermore, table 1 could present the $N_2O$ fluxes as well as the concentrations.

Ln 366 - 369: Figure 4a should be mentioned, where the sampling stations considered in this study are shown in detail.

**Figures:**

Figures 2 and 3 and Supplemetary figures 3 and 5. It is not necessary to write the word "concentration" on the axes of the figures when referring to $N_2O$ concentration (nM), just as you do not write $NO_x^-$ concentration.

**References**

**Ln 604-613:** Rosentreter references should be put in chronological order.

---

## Referee Report (RR2)

I thank the authors for thoroughly addressing the first round of reviews. In the revised version they discussed a possible dilution effect of WWTPs effluents, included several parametrizations for the gas transfer coefficient, addressed seasonal effects and added missing information about their sampling strategies. I have no further comments and suggest the paper for publication.

---

## Author Response (AR2)

We thank the reviewers for commenting on the manuscript, which helped to further improve the manuscript. We have responded to the reviewers' comments in blue below and applied changes accordingly in the manuscript.

The paper by Tang et al., with the title: "Variable contribution of wastewater treatment plant effluents to N2O emission " has greatly improved in clarity and quality with the new changes. Given that there are very few studies worldwide on the impact of WWTPs on N2O emissions in aquatic systems, as shown in the Supplementary figure 8 and in the Table 1 of the paper, I consider the article to be of great interest and that it will be widely disseminated. However, I have some suggestions:

General comments:
The title of the paper refers to the contribution of WWTPs to N2O emissions. However, the manuscript hardly discusses water-atmosphere fluxes of N2O, nor the contribution of WWTPs to water-atmosphere fluxes of N2O in the Potomac River estuary in detail. So my suggestion is that just as the contribution of WWTPs to N2O concentration is discussed, there should be more discussion of the effect of WWTPs on N2O fluxes to the atmosphere in the system. Another option would be to change the title of the paper.

Following the suggestion from the reviewer, we added more discussion about the influence of WWTPs on $N_2O$ emissions along with the discussion about $N_2O$ concentrations (see detailed responses below). In addition, we changed the title to "Variable contribution of wastewater treatment plant effluents to downstream nitrous oxide concentrations and emissions".

Material and methods:
Ln 158: There is no mention of phosphate in the manuscript, so you should delete it. Were the samples taken in triplicate like the N2O samples? Please indicate.

Since we used phosphorus in the correlation analysis of Figure 3, we decided to keep the description of phosphorus measurements. One sample was collected for phosphorus measurements. This has been updated in the manuscript.

Ln 167-168: Were obtained N2O concentrations in the water from the measurements made in the headspace using the solubility proposed by Weiss and Price (1980)?

We were not measuring $N_2O$ concentrations in the headspace. Instead, we measured the total amount of $N_2O$ dissolved in the water by a mass spectrometer. The $N_2O$ concentration was then calculated by dividing the amount of $N_2O$ in the water by the water volume. This is detailed in lines 164-172.

Ln 168-170: The text: "Specifically, the monthly atmospheric N2O concentrations were obtained from the nearby atmospheric station in Brentwood, Maryland (https://gml.noaa.gov/) (Andrews et al., 2023)." should be included in the N2O flux calculation section, it could go on line 191 after (Weiss and Price, 1980).

Following the reviewer's suggestion, we moved this sentence after (Weiss and Price, 1980) to describe how equilibrium $N_2O$ concentration was calculated in lines 177-179.

Ln 173: It should be included how you have calculated the saturation percentage. This parameter is discussed in the text and presented in figure 2a.

We have added the equation to calculate $N_2O$ saturation: $N_2O$ saturation (%) is calculated: $\frac{N_2O_{measured}}{N_2O_{equilibrium}} \times 100$.

Ln 186: It should be indicated how the total N cited in lines 158-160 has been measured.

We have clarified how total N was estimated in lines 160-161: "Total N is the sum of total Kjeldahl nitrogen and nitrite plus nitrate."

Ln 191-194. The three gas transfer velocity (k) equations should be written in the same format:
- Or write the k proposed by Wanninkhof (2014) as k660 as has been done for the other two parameterisations.:
k_660=0.251 x U^2
- Or write:
Raymond and Cole (2001): k=1.91 x e^0.35xU x (Sc/600)^(-0.5)
Jiang et al. (2008): k=0.314 x U^2-0.436 x U+3.99 x (Sc/600)^(-0.5)
Wanninkhof (2014): k=0.251 x U^2 x (Sc/660)^(-0.5)

The three gas transfer velocity equations are now written in the same format based on the reviewer's suggestion.

Results and discussion
Ln 219-228. More should be commented on the water-atmosphere N2O fluxes, practically only their range of variation in the whole system is presented. As with the N2O concentrations, the water-atmosphere fluxes present seasonal variations (this if is commented in the abstract) and surely present spatial variations (you should comment on this). However, it is mentioned in the paper that the saturation percentage of N2O is always higher than 100%, so the system behaves as a source of this gas, and that there is seasonal variation, but little is said about the fluxes to the atmosphere (Ln 218-220).

Following the reviewer's suggestion, we have added more discussion about $N_2O$ fluxes/emissions: "$N_2O$ fluxes ranged from 1 to 31.7 µmol $N_2O$ m$^{-2}$ d$^{-1}$, generally decreasing from upstream to downstream (Figure 2d). $N_2O$ fluxes showed a similar seasonal pattern to $N_2O$ saturation: higher in summer and fall."

Ln 226-228: I do not believe that a maximum flux of N2O to the atmosphere of 31.7µmol m-2 d-1 in the Potomac River Estuary can be considered as an intense source of N2O to the

atmosphere, as there are other estuaries with much more intense emissions. Perhaps it would be more accurate to put: Therefore, tributaries to the Chesapeake Bay (i.e., the Potomac River) are more intense sources of N2O to the atmosphere than the Bay.

We have modified the sentence as the reviewer suggested: "Therefore, the tributaries (i.e., Potomac River) are more intense sources of $N_2O$ to the atmosphere than mainstem of the bay".

Ln 285-286: For better clarity and interpretation of the text, the values of your observed $\delta$15N of N2O downstream of WWTPs and in the urban WWTPs should be included.

Since we have listed $\delta^{15}N$ of $N_2O$ downstream of WWTPs and in the urban WWTPs in the text above, to avoid repeating, we modified this sentence to: "The $\delta^{15}N$ values of $N_2O$ in these urban WWTPs were lower than those found in waters downstream of WWTPs in the Potomac River (median $\delta^{15}N$ at 13‰)".

Ln 288: Do you know of any work where there is evidence of denitrification in WWTPs, in downstream creeks, or in sediments? If so, could you please cite it.

$N_2O$ production from denitrification has been observed in WWTPs, creeks and sediments (e.g., Toyoda et al., 2011; Beaulieu et al., 2011). We hypothesized the partial $N_2O$ reduction caused the elevated $\delta^{15}N$ of $N_2O$ downstream of WWTPs. This partial $N_2O$ reduction has been suggested to explain the occurrence of high $\delta^{15}N$ of $N_2O$ in the core of marine oxygen minimum zones (Kelly et al., 2011; Bourbonnais et al., 2017). However, the effect of denitrification, especially the partial $N_2O$ reduction to $N_2$, on $N_2O$ isotopes in WWTPs, creeks and sediments remain to be evaluated.

Ln 343-345: Text in brackets is not in Times New Roman 12. Why is the number of data considered for the predictions so small? Especially for the stations without WWTPs, in the complete study there are 8 sampling x 8 stations (4 stations without WWTPs + 4 stations central channel) = 64 data compared to the 23 considered in the prediction.

The font in the equation (Cambria Math) is different from the font in the text (Times New Roman 12). The lower number of stations used for prediction is because not all the stations were sampled during each field campaign or not all the parameters of each station were measured. See the Dataset deposited in Zenodo repository for the sampled stations and measured parameters.

Ln 365. In the section: "Impact of wastewater treatment plants on N2O concentrations and emissions" very little is mentioned about how N2O fluxes to the atmosphere vary in the stations upstream and downstream of the WWTPs. However, there is much discussion of the effect of the WWTPs on N2O concentrations. More should be said about these emissions, as the title of the paper says "Variable contribution of wastewater treatment plant effluents to N2O emission". Furthermore, table 1 could present the N2O fluxes as well as the concentrations.

Following the reviewer's suggestion, we have added more description of the $N_2O$ fluxes/emissions in this section.

"Interestingly, the $N_2O$ concentration and flux at the station downstream of Mooney WWTP in Neabsco Creek were lower than the $N_2O$ concentration and flux at the station upstream of Mooney WWTP (15.0 nM vs 20.1 nM; 14.6 µmol m$^{-2}$ d$^{-1}$ vs 24.7 µmol m$^{-2}$ d$^{-1}$).".

"In contrast, we found a substantially higher $N_2O$ concentration and flux downstream of the Noman Cole WWTP than the upstream station (30.8 nM vs 16.7 nM; 55 µmol m$^{-2}$ d$^{-1}$ vs 17.6 µmol m$^{-2}$ d$^{-1}$) in the Pohick Creek, which is less affected by the tidal cycle due to its semi-closed geography (salinity was 0.12)."

Table 1 shows either $N_2O$ concentrations or fluxes or both depending on which one is available from previous studies. We have added $N_2O$ fluxes from our study to the table.

Ln 366 - 369: Figure 4a should be mentioned, where the sampling stations considered in this study are shown in detail.

We have added Figure 4a to the end of this sentence.

Figures:
Figures 2 and 3 and Supplemetary figures 3 and 5. It is not necessary to write the word "concentration" on the axes of the figures when referring to N2O concentration (nM), just as you do not write NOx- concentration.

Based on the reviewer's suggestion, we used $N_2O$ (nM) when referring to $N_2O$ concentration in the figure axes titles.

References
Ln 604-613: Rosentreter references should be put in chronological order.

Rosentreter references are now cited in chronological order in the reference list.

References:

Beaulieu, J. J., Tank, J. L., Hamilton, S. K., Wollheim, W. M., Hall, R. O., Jr., Mulholland, P. J., Peterson, B. J., Ashkenas, L. R., Cooper, L. W., Dahm, C. N., Dodds, W. K., Grimm, N. B., Johnson, S. L., McDowell, W. H., Poole, G. C., Valett, H. M., Arango, C. P., Bernot, M. J., Burgin, A. J., Crenshaw, C. L., Helton, A. M., Johnson, L. T., O'Brien, J. M., Potter, J. D., Sheibley, R. W., Sobota, D. J., and Thomas, S. M.: Nitrous oxide emission from denitrification in stream and river networks, Proceedings of the National Academy of Sciences of the United States of America, 108, 214-219, 10.1073/pnas.1011464108, 2011.

Bourbonnais, A., Letscher, R. T., Bange, H. W., Echevin, V., Larkum, J., Mohn, J., ... & Altabet, M. A.: $N_2O$ production and consumption from stable isotopic and concentration data in the Peruvian coastal upwelling system, Global Biogeochemical Cycles, 31(4), 678-698, 2017.

Kelly, C. L., Travis, N. M., Baya, P. A., and Casciotti, K. L.: Quantifying Nitrous Oxide Cycling Regimes in the Eastern Tropical North Pacific Ocean With Isotopomer Analysis, Global Biogeochemical Cycles, 35, 10.1029/2020gb006637, 2021.

Toyoda, S., Suzuki, Y., Hattori, S., Yamada, K., Fujii, A., Yoshida, N., Kouno, R., Murayama, K., and Shiomi, H.: Isotopomer Analysis of Production and Consumption Mechanisms of $N_2O$ and $CH_4$ in an Advanced Wastewater Treatment System, Environmental Science & Technology, 45, 917-922, 10.1021/es102985u, 2011.